# Long time series of daily evapotranspiration in China based on the SEBAL model and multisource images and validation

**Minghan Cheng[1,2,3], Xiyun Jiao[2,3*], Binbin Li[4], Xun Yu[1], Mingchao Shao[1], Xiuliang Jin[1*]**

[1]Institute of Crop Sciences, Chinese Academy of Agricultural Sciences/Key Laboratory of Crop Physiology and Ecology, Ministry of Agriculture, Beijing 100081, PR China

[2]Hohai University, College of Agricultural Science and Engineering, Nanjing, Jiangsu Province, 210048, PR China

[3]State Key Laboratory of Hydrology-Water Resources and Hydraulic Engineering, Nanjing, Jiangsu Province, 210048, PR China

[4]Monitoring Center of Soil and Water Conservation, Ministry of Water Resources of the People's Republic of China, Beijing, 100053, PR China

*Correspondence to*: Xiyun Jiao (xyjiao@hhu.edu.cn) and Xiuliang Jin (jinxiuliang@caas.cn)

**Abstract.** Satellite observations of evapotranspiration (ET) have been widely used for water resources management in China. An accurate ET product with a high spatiotemporal resolution is required for research on drought stress and water resources management. However, such a product is currently lacking. Moreover, the performances of different ET estimation algorithms for China have not been clearly studied, especially under different environmental conditions. Therefore, the aims of this study were as follows: (1) to use multisource images to generate a long time series (2001-2018) daily ET product with a spatial resolution of 1 km × 1 km based on the Surface Energy Balance Algorithm for Land (SEBAL); (2) to comprehensively evaluate the performance of the SEBAL ET in China using flux observational data and hydrological observational data; (3) to compare the performance of the SEBAL ET with the MOD16 ET product at the point-scale and basin-scale under different environmental conditions in China. At the point-scale, both the models performed best in the conditions of forest cover, subtropical zones, hilly terrain, or summer, respectively, and SEBAL performed better in most conditions. In general, the accuracy of the SEBAL ET (rRMSE = 44.91%) was slightly higher than that of the MOD16 ET (rRMSE = 48.72%). In the basin-scale validation, both the models performed better than in the point-scale validation, with SEBAL obtaining superior results (rRMSE = 13.57%) to MOD16 (rRMSE = 32.84%). Additionally, both the models showed a negative bias, with the bias of the MOD16 ET being higher than that of the SEBAL ET. In the daily-scale validation, the SEBAL ET product showed an RMSE of 0.92 mm/d and an r-value of 0.79. In general, the SEBAL ET product can be used for the qualitative analysis and most quantitative analysis of regional ET. SEBAL ET product is freely available at https://doi.org/10.5281/zenodo.4243988 (Cheng, 2020). The results of this study can provide a reference for the application of remotely sensed ET products and the improvement of satellite ET observation algorithms.

**Keyword:** evapotranspiration; SEBAL; MOD16; accuracy validation; multiscale

**1. Introduction**

Evapotranspiration (ET) is the process of transferring surface water to the atmosphere, including soil evaporation and vegetation transpiration (Wang and Dickinson, 2012). This process is a key node linking surface water and energy balance. In the process of water balance, ET represents the consumption of surface water resources, and in the process of energy balance, the energy consumed by ET is called the latent heat flux ($\lambda$ET, W/m$^2$, where $\lambda$ is the latent heat vaporization), which is an important energy component (Helbig et al., 2020; Zhao et al., 2019). Approximately 60% of global precipitation ultimately returns to the atmosphere through evapotranspiration (Wang and Dickinson, 2012). Therefore, accurately quantifying the ET of different land cover types is necessary to better understand changes in regional water resources. However, the methods for the estimation of ET based on point-scale or small-area-scale analysis, such as lysimeter and eddy covariance, cannot meet the requirement of global climate change research and regional water resource management (Li et al., 2018). Since the United States successfully launched the first meteorological satellite in the 1960s, hydrological remote sensing (RS) applications have developed rapidly and have led to huge breakthroughs (Karimi and Bastiaanssen, 2015). Remote sensing technology with a high spatiotemporal continuity provides an effective means for regional ET estimation.

Satellite remote sensing provides a reliable direct estimation of ground parameters; however, it cannot measure ET directly (Wang and Dickinson, 2012). Therefore, several RS-based algorithms for the estimation of ET have been proposed and reviewed (Pôças et al., 2020; Senay et al., 2020; Wang and Dickinson, 2012). These models can be divided into three types according to their mechanism: those based on surface energy balance residual (SEBR), those based on semi-empirical formulas (SEFs) and statistic methods. SEBR-based models can be further divided into one-source models and two-source models (Wang and Dickinson, 2012). One-source models do not distinguish vegetation from bare soil and regard the land surface as a system that exchanges energy and water with the atmosphere. Examples of one-source models include the Surface Energy Balance Index (S-SEBI) (Roerink et al., 2000), the Surface Energy Balance System (SEBS) (Su, 1999), and the Surface Energy Balance Algorithm for Land (SEBAL) (Bastiaanssen et al., 1998a; Bastiaanssen et al., 1998b). These models have a theoretical basis, a simple principle, strong portability, and have been widely used (Bastiaanssen and Steduto, 2017; Elnmer et al., 2019; Huang et al., 2015; Wagle et al., 2019). Two-source models distinguish the surface water and energy exchange between vegetation and bar soil and calculate fractional canopy coverage (*Fc*) using an empirical formula and a vegetation index obtained from remote sensing data to divide the land surface into vegetation and bare soil in each single pixel. Examples of two-source models include the Two-source Energy Balance (TSEB) (Kustas et al., 2003), Two-source Trapezoid Model for Evapotranspiration (TTME) (Long and Singh, 2012), and Hybrid Dual-source Scheme and Trapezoid Framework-based Evapotranspiration Model (HTEM) (Yang and Shang, 2013). Compared to one-source models, two-source models have a superior theoretical mechanism. SEF-based models using traditional semi-empirical formulas calculate $\lambda$ET and are simpler than SEBR-based models. Examples of SEF-based models include the Surface Temperature and Vegetation Index

(T$_s$-VI) space model (Carlson, 2007) and the Global Land Evaporation Amsterdam Model (GLEAM) based on the Priestley–Taylor (P-T) equation (Miralles et al., 2011). Another well-known SEF-based model is based on the Penman–Monteith (P-M) equation, which has been improved and applied to remote sensing data to estimate regional ET (Mu et al., 2007; Mu et al., 2011). Moreover, statistic methods for ET estimation by using statistical regression or machine learning to fit multiple indicators (e.g., meteorological data or remote sensing data) and in situ ET are also be widely used (Mosre and Suárez, 2021; Yamaç and Todorovic, 2020).

Since ET plays a critical role in the study of hydrology and ecology, ET products with a high spatiotemporal resolution are required. Therefore, a growing number of ET products have been generated to meet research needs. These include MOD16, which is generated by NASA based on the Penman-Monteith algorithm and has a spatial resolution of 500 m × 500 m and a temporal resolution of eight days (Mu et al., 2007; Mu et al., 2011). The GLEAM daily ET product with a spatial resolution of 0.25° × 0.25° has been generated by the University of Bristol, UK, based on the Priestley-Taylor method (Miralles et al., 2010). Additionally, Chen generated long time series daily ET datasets with a spatial resolution of 0.1° × 0.1° based on the SEBS algorithm (Chen et al., 2014a; Chen, 2019). However, there are few ET products which simultaneously meet the current research needs in terms of temporal and spatial resolution. Therefore, generating a kilometer-level daily ET product which can minimize the influence of mixed pixels is critical.

Water resources management is essential for China as it has an unbalanced spatial and temporal distribution of water resources. ET, as a crucial component of terrestrial water cycle, is critical for understanding water resources budget in China. Therefore, a spatiotemporal continuous ET data is needed. Several studies evaluated the performance of various remote sensing-based algorithm in China. For example, Chen et al. (2014b) used 23 eddy covariance (EC) sites to evaluate the performance of Penman-Monteith method (used for generating MOD16 product) and Priestley-Taylor method (used for generating GLEAM product) in China, however, these model can only explained approximately 61%-80% of the variability in ET. Li et al. (2017) used a SEBR-based model – SEBS to map the ET in Heihe River Basin, Northwest China and evaluated its performance in different land cover types, in general, SEBS outperformed than Priestley-Taylor method, but SEBS showed significant bias in several land cover types, e.g., village (mainly croplands). Sun et al. (2020) evaluated the performance of Shuttleworth–Wallace–Hu (SWH) and SEBAL in Northwest China and showed better accuracy than MOD16 product. In general, the accuracy of ET derived from satellite imagery is affected by spatiotemporal conditions (Wagle et al., 2017). Several studies have indicated that RS-based methods for modeling ET have errors of 15–50% (Velpuri et al., 2013; Xue et al., 2020). RS-based models have different applicable conditions, and understanding the variation in accuracy between such models is important for their reasonable application. However, few studies have validated the robustness of different models using long time series and at a large spatial scale. For China with large area and complex terrain, few studies clearly discussed the performance of RS-based models under different environmental conditions, most

studies only aimed at a certain area. Moreover, there is no ET product for China with a high spatiotemporal resolution, and the applicability of different RS-based models for the estimation of ET in China is not clear, which hampers the management of ET.

In order to improve ET products in China and better understand the performance of RS-based ET estimation models in China, in this paper, we aim to (1) generate a long time series daily ET product with a spatial resolution of 1 km × 1 km based on the SEBAL model and multisource remote sensing images, (2) validate the accuracy of the generated ET product in China based on flux tower observational data and hydrological data, and (3) compare the performance of the generated ET product with MOD16 datasets in China under different environmental conditions.

## 2. Materials and methods

### 2.1 Study area

China ($3°31'00''$–$53°33'47''$ N, $73°29'59.79''$–$135°2'30''$ E) covers a land area of approximately 9,600,000 km$^2$, mainly including temperate zones, warm-temperate zones, subtropical zones, tropical zones, and plateau climate zones. China can be divided into nine basin regions based on the distribution of water resources (Zhang et al., 2011): the Southwest Basin (SwB), Continental Basin (CB), Pearl River Basin (PRB), Yangtze River Basin (YRB), Southeast Basin (SeB), Haihe River Basin (HRB), Yellow River Basin (YeRB), Huaihe River Basin (HuRB) and Songhua and Liaohe River Basin (SLRB) (Fig. 1).

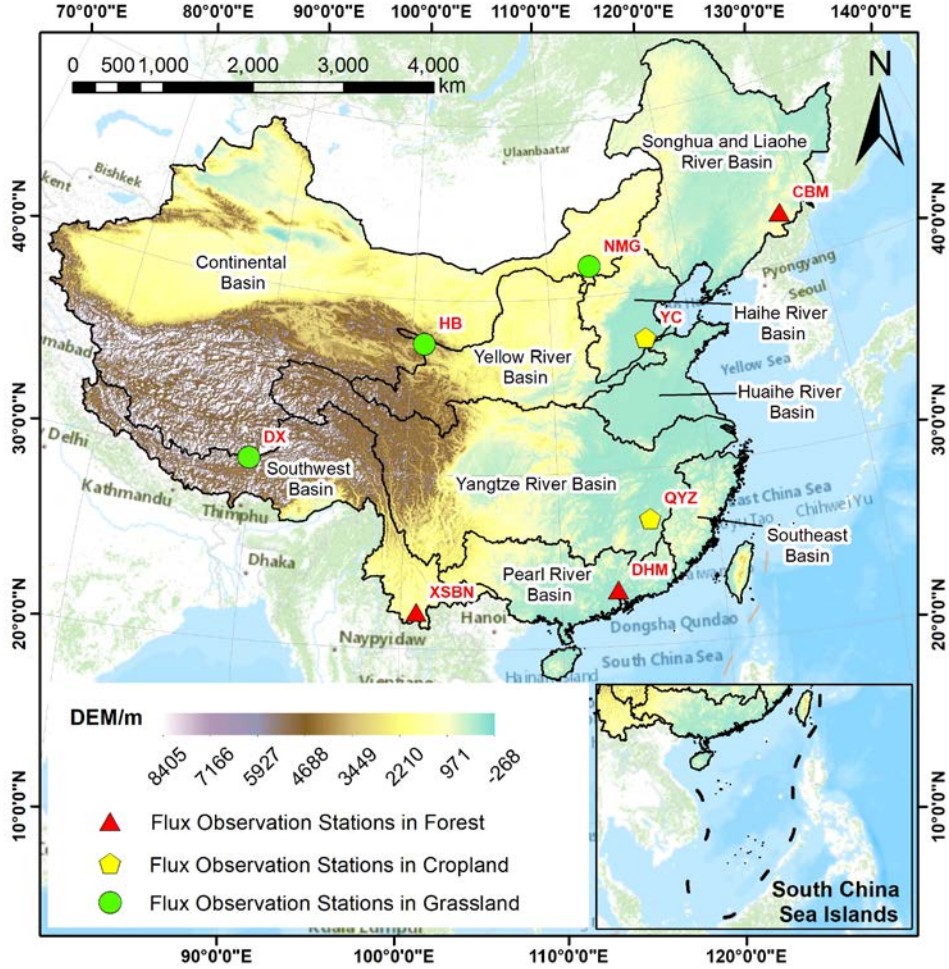

**Figure 1.** The location of the study area. Note: CBM: Changbai mountain; DHM: Dinghu mountain; DX: Dangxiong; HB: Haibei; NMG: Neimenggu; QYZ: Qianyanzhou; XSBN: Xishuangbanna; YC: Yucheng. (Note: the Chinese boundry was obtained from Institute of Geographic Science and Natural Resources Research, Chinese Academy of Sciences (http://www.resdc.cn/))

**2.2 Generation of long time series daily ET product**

In this study, a long time series daily ET product was generated based on SEBAL, which is a widely used one-source model (Gobbo et al., 2019; Jaafar and Ahmad, 2020; Mhawej et al., 2020; Rahimzadegan and Janani, 2019). SEBAL has been shown to have a good performance for ET estimation and can be regarded as typical of SEBR-based models (Bastiaanssen et al., 1998b; Timmermans et al., 2006; Wagle et al., 2017). The workflow for the calculation of the daily ET using the SEBAL model and multisource satellite images is shown in Fig. 2. The SEBAL model calculates the instantaneous λET of the satellite transit time as a residual based on the surface energy balance equation (Eq. 1) as follows:

$$\lambda ET = R_n - G - H \tag{1}$$

where $R_n$ is the net radiation flux, $H$ is the sensible heat flux, and $G$ is the soil heat flux (the unit of all three parameters is W/m$^2$).

In this paper, MODIS data (MCD43 surface albedo, MOD11 surface temperature (daytime), MOD13 NDVI) and meteorological data (air temperature) from the Global Modeling and Assimilation Office (GMAO) were used as input for surface parameterization ($R_n$, G and H). The details of generating SEBAL ET can be referred to Appendix.

The spatial and temporal resolutions of the MCD43 surface albedo and the MOD11 daytime surface temperature are 1 day and 1 km × 1 km, while those of MOD13 NDVI are 16 days and 500 m × 500 m. In this study, MOD13 was resampled to 1 km × 1 km and processed by smoothing and gap-filling from time series to daily data (Vuolo et al., 2017). It should be noted that there are several missing or unreliable pixels in MODIS images which may cause by cloud or other reasons, these pixels were marked in

quality control (QC) files. In this study, these anomalous pixels of MODIS dataset (MOD11, MOD13 and MCD43) were filled referred to previous studies, the rules as follows: (1) the value of anomalous pixel will be computed by liner interpolation of the nearest reliable value after it or prior it; (2) if the anomalous pixel was found in the first or last day, it will be replaced by the closest reliable date value. The more details can be referred to the study of Mu et al. (2011) and Zhao et al. (2005). Land surface temperature is the crucial parameter for SEBAL ET generating, the Appendix 2 shows the ratio of interpolated pixels of land surface temperature

(MOD11) data (31% ± 11%). The spatial and temporal resolutions of GMAO air temperature are 1 day and 0.25° × 0.25°, respectively. The coarse-resolution GMAO data were non-linearly interpolated to a spatial resolution of 1 km × 1 km based on the four GMAO pixels surrounding a given pixel (Zhao et al., 2005). The spatial and temporal resolutions of wind speed are 1 day and 1 km × 1 km (China Meteorological Data Network, http://data.cma.cn). The final generated daily ET product has a spatial resolution of 1 km × 1 km and covers the period 2001 to 2018.

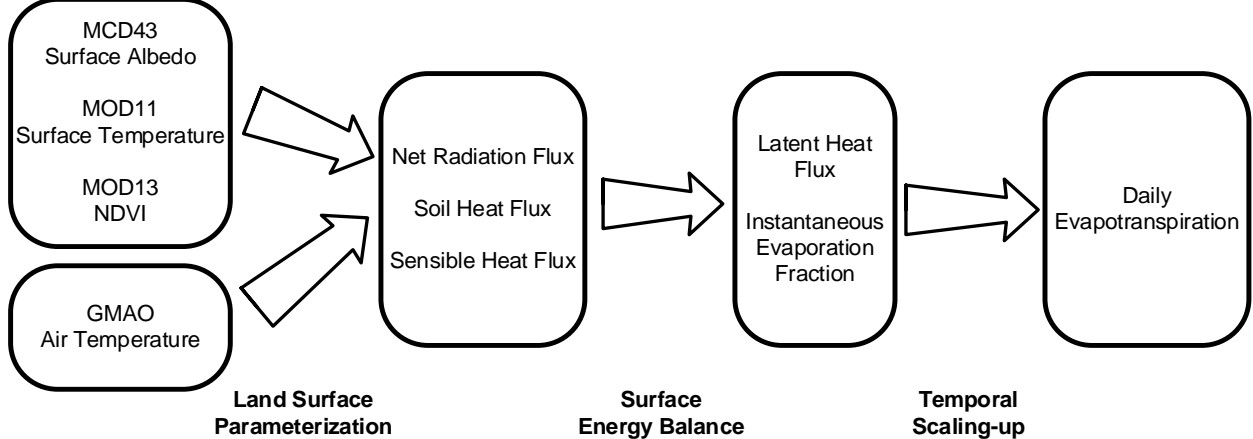

**Figure 2.** A flowchart of the Surface Energy Balance Algorithm for Land (SEBAL) which was used to convert multisource images to daily evapotranspiration.

**2.3 Validation methods**

**2.3.1 Point-scale validation**

The eddy covariance method measures $\lambda ET$ from the covariance between moisture fluxes and vertical wind velocity using rapid response sensors at frequencies typically equal to or greater than 10 Hz (Wang and Dickinson, 2012); it is regarded as the most effective method for the estimation of ET and has been widely used (Wang and Dickinson, 2012). In this study, eddy covariance tower-measured daily flux data from eight stations in China (Table 1) obtained in 2003–2010 were used to validate the modeled ET ($ET_{SEBAL}$, $ET_{MOD}$). The latent heat flux ($\lambda ET$) observed at the flux towers was converted into the observed ET ($ET_{flux}$). It should be noted that the energy balance closure issue, which indicates the sum of sensible heat (H), latent heat ($\lambda ET$) and soil heat flux (G) is not equal to net radiation (Rn), was often found in eddy covariance system. Therefore, the eddy covariance system measured value should be filtered and corrected. First, the data with Energy Balance Closure Ratio (ECR, Eq. 2) less than 80% were not selected for validation (Wang et al., 2019), and then, the remaining data with ECR more than 80% were corrected by using Bowen Ratio energy balance correction (Eq. 3) (Chen et al., 2014b).

$$ECR = \frac{H + \lambda ET}{Rn - G} \tag{2}$$

$$\lambda ET_{cor} = \frac{Rn - G}{H + \lambda ET} \times \lambda ET \tag{3}$$

where Rn, G, H and $\lambda ET$ are all eddy covariance system measured value, and $\lambda ET_{cor}$ is corrected value. To ensure a reliable evaluation, the pixel value where the flux tower located (area of 1 km × 1 km) was extracted for comparison with measured value (Velpuri et al., 2013). The water demand is different under different environmental conditions. Therefore, it is necessary to understand the accuracy performance of ET products for different vegetation types when a single ET product is not comprehensive (Velpuri et al., 2013). In order to better understand the influence of different environmental conditions on the accuracy of the model, the modeled ET were validated for different terrain, climate zones, land cover types, and seasons. Additionally, MOD16 data were resampled to a spatial resolution of 1 km × 1 km and daily $ET_{SEBAL}$ and daily $ET_{flux}$ data were accumulated to eight days to match the MOD16 data. $ET_{SEBAL}$ was validated at the daily scale and 8-day scale, respectively.

**Table 1.** Details of the eight flux observation stations.

| Station | Observation period | Longitude | Latitude | Elevation | Ecosystem types | Climate Zone |
|---|---|---|---|---|---|---|
| Changbai mountain (CBM) | 2003–2010 | 128.10 | 42.40 | 738 | Forest | temperate zone |
| Dinghu mountain (DHM) | 2003–2010 | 112.53 | 23.17 | 300 | Forest | subtropical zone |
| Dangxiong (DX) | 2004–2010 | 91.07 | 30.85 | 4333 | Grassland | plateau climate zone |
| Haibei (HB) | 2003–2010 | 101.29 | 37.62 | 3250 | Grassland | plateau climate zone |
| Neimenggu (NMG) | 2004–2010 | 116.68 | 43.55 | 1200 | Grassland | temperate zone |
| Qianyanzhou (QYZ) | 2003–2010 | 115.06 | 26.74 | 102 | Forest | subtropical zone |
| Xishuangbanna (XSBN) | 2003–2010 | 101.20 | 21.96 | 750 | Forest | tropical zone |
| Yucheng (YC) | 2003–2010 | 116.60 | 36.95 | 28 | Cropland | warm-temperate zone |

### 2.3.2 Regional-scale validation

Furthermore, the regional (basin-scale) ET was calculated using the water balance method (Eq. 34) to validate the modeled ET at the regional scale.

$$ET = P - Q - \Delta S \tag{4}$$

where $P$ (unit: mm) is the annual precipitation in the basin; $Q$ (unit: mm) is the annual runoff in the basin, which includes surface runoff and groundwater runoff; $\Delta S$ is the change in the groundwater and surface water storage in a year; the change of $\Delta S$ over multiple years can be ignored (Liu et al., 2016; Senay et al., 2011). The average ET over multiple years was calculated in each primary water resources division in China (the nine basins shown in Fig. 1) from 2001 to 2018; these values of ET are referred to as $ET_{WB}$.

### 2.3.3 Accuracy estimation

The modeled ET values were compared with the observed ET ($ET_{flux}$, $ET_{WB}$) to evaluate the performance of $ET_{SEBAL}$ and $ET_{MOD}$, respectively. The correlation coefficient (r), root-mean-square error (RMSE), relative root-mean-square error (rRMSE), and mean bias error (MBE) were selected to quantify the accuracy of the modeled ET. The equations for these parameters are shown below:

$$r = \frac{\sum_{i=1}^{n}(ET_{Mi} - \overline{ET_M})(ET_{Obi} - \overline{ET_{Ob}})}{\sqrt{\sum_{i=1}^{n}(ET_{Mi} - \overline{ET_M})^2 \sum_{i=1}^{n}(ET_{Obi} - \overline{ET_{Ob}})^2}} \tag{5}$$

$$RMSE = \sqrt{\frac{1}{n}\sum_{i=1}^{n}(ET_{Mi} - ET_{Obi})^2} \tag{6}$$

$$rRMSE = \frac{RMSE}{\overline{ET_{Ob}}} \times 100\% \tag{7}$$

$$MBE = \frac{1}{n}\sum_{i=1}^{n}(ET_{Mi} - ET_{Obi}) \tag{8}$$

where $ET_M$ is the modeled ET ($ET_{SEBAL}$ and $ET_{MOD}$); $ET_{Ob}$ is the observed ET ($ET_{flux}$ and $ET_{WB}$); and $n$ is the number of samples. r was calculated to evaluate the linear relationship between the modeled and observed ET; higher r-values mean a higher correlation. RMSE and rRMSE were used to evaluate the performance of the model: smaller RMSE and rRMSE mean a higher accuracy. rRMSE is a critical indicator to evaluate the accuracy of a model (Jin et al., 2020). The MBE was used to measure whether the result was overestimated (positive values of MBE) or underestimated (negative values of MBE).

### 2.4 Data sources and tools used

#### 2.4.1 MOD16 data

The MOD16 ET product is one of widely used evapotranspiration dataset for water resources management and global change study, which also performs accurate to some extent (He et al., 2019; Mu et al., 2011). In this study, the comparison of SEBAL ET and MOD16 ET was conducted to judge if the further improvement was found in SEBAL ET. The MOD16 ET data ($ET_{MOD}$) were produced using an ET algorithm based on the P-M equation (Eq. 9) (Monteith, 1965) that has been improved (Mu et al., 2007; Mu et al., 2011).

$$\lambda ET = \frac{sA + \rho C_p VPD / r_a}{s + \gamma(1 + r_s / r_a)} \tag{9}$$

where $s$ (unit: Pa/K) is the slope of the temperature-saturated water pressure curve at the current temperature; $A$ (unit: W/m$^2$) is the available energy; $\rho$ (unit: kg/m$^3$) is the air density; $C_p$ (unit: J/(kg×K)) is the specific heat of air at constant pressure; $VPD$ (unit: Pa) is the difference in water vapor pressure; $\gamma$ (unit: Pa/K) is the psychrometric constant; and $r_a$ and $r_s$ (unit: s/m) are the aerodynamic resistance and surface resistance, respectively. The MOD16 ET data are available for regular 500 m grid cells for the entire global

vegetated land surface at 8-day composite, and the data do not cover regions corresponding to water, barren land, and buildings (He et al., 2019). In this study, MOD16 data were obtained from the NASA Atmosphere Archive & Distribution System Distributed Active Archive Center (LAADS DAAC, https://ladsweb.modaps.eosdis.nasa.gov).

#### 2.4.2 Auxiliary data

In order to ensure the objectivity of the comparison between the SEBAL and P-M models, MODIS satellite data were selected as the input for SEBAL, including the surface albedo (MCD43), surface temperature (MOD11), and NDVI (MOD13) obtained from LAADS DAAC. Additionally, gridded air temperature data were obtained from the GMAO (https://gmao.gsfc.nasa.gov). Flux-tower observational data were obtained from ChinaFLUX (www.chinaflux.org). Precipitation and runoff data for each basin from 2001 to 2018 were obtained from the Water Resources Bulletin provided by the Ministry of Water Resources of the People's Republic of

China (http://www.mwr.gov.cn/).

#### 2.4.3 Tools used

Python (version 3.7; Google Inc., Mountain View, California, USA) and the Geospatial Data Abstraction Library (GDAL; version 3.1.1; Google Inc.) were used to construct SEBAL. The ArcGIS software (version 10.4; Esri Inc., Redlands, California, USA) and

ENVI software (version 5.3; Esri Inc.) were used to process raster data. Python and the SPSS software (version 21; IBM Inc., Armonk, New York, USA) were used for numerical calculation and analysis.

## 3. Results

### 3.1 Validation of daily SEBAL ET at the point-scale using flux tower observations

The validation results for the daily SEBAL ET ($ET_{SEBAL}$) obtained using flux tower observational data are shown in Fig. 3. Compared to $ET_{flux}$, $ET_{SEBAL}$ showed a good performance in China; the two data showed a high consistency, with an r-value of 0.79 with 9896 samples. However, the bias of SEBAL was relatively high; the RMSE and rRMSE were 0.92 mm/d and 42.04%, respectively. As shown in the scatter diagrams in Fig. 3, $ET_{SEBAL}$ showed a negative bias at high values and a positive bias at low values. In general, SEBAL underestimated ET in China, with an MBE of –0.15 mm/d. Moreover, the daily $ET_{SEBAL}$ performed similarly for different land use types. The daily $ET_{SEBAL}$ had a bias of 0.95 mm/d (rRMSE = 37.24%) in cropland and 0.89 mm/d (rRMSE = 44.25%) in grassland, and the daily $ET_{SEBAL}$ underestimated in both cropland and grassland, with MBEs of –0.26 mm/d and –0.44 mm/d, respectively. In forest, the daily $ET_{SEBAL}$ had the highest RMSE of 1.02 mm/d (rRMSE = 41.25%) and the lowest r-value of 0.73, and slightly overestimated compared to $ET_{flux}$ (MBE = 0.09 mm/d). Fig. 4 showed the time series variation of ET. In general, SEBAL ET and observed ET both showed a clear seasonal variation characteristic among the eight flux tower stations. Moreover, an annual periodic variation was found in most stations (Figs. 4a, b, c, d, e and g). In which, the cropland stations (YC and QYZ) presented a relatively disordered period (Figs. 4 g and h). Which was likely contributed to the double-crop rotation system used in these regions. For example, YC station which located in North China Plain, generally was maize and wheat rotation, which may cause two peaks of crop water consumption (ET) occurred in one year. Furthermore, Figs. 3 and 4 both showed that SEBAL ET was clearly underestimated at higher ET rates. The observed ET fluctuated higher than SEBAL ET in all stations as shown in Fig. 4.

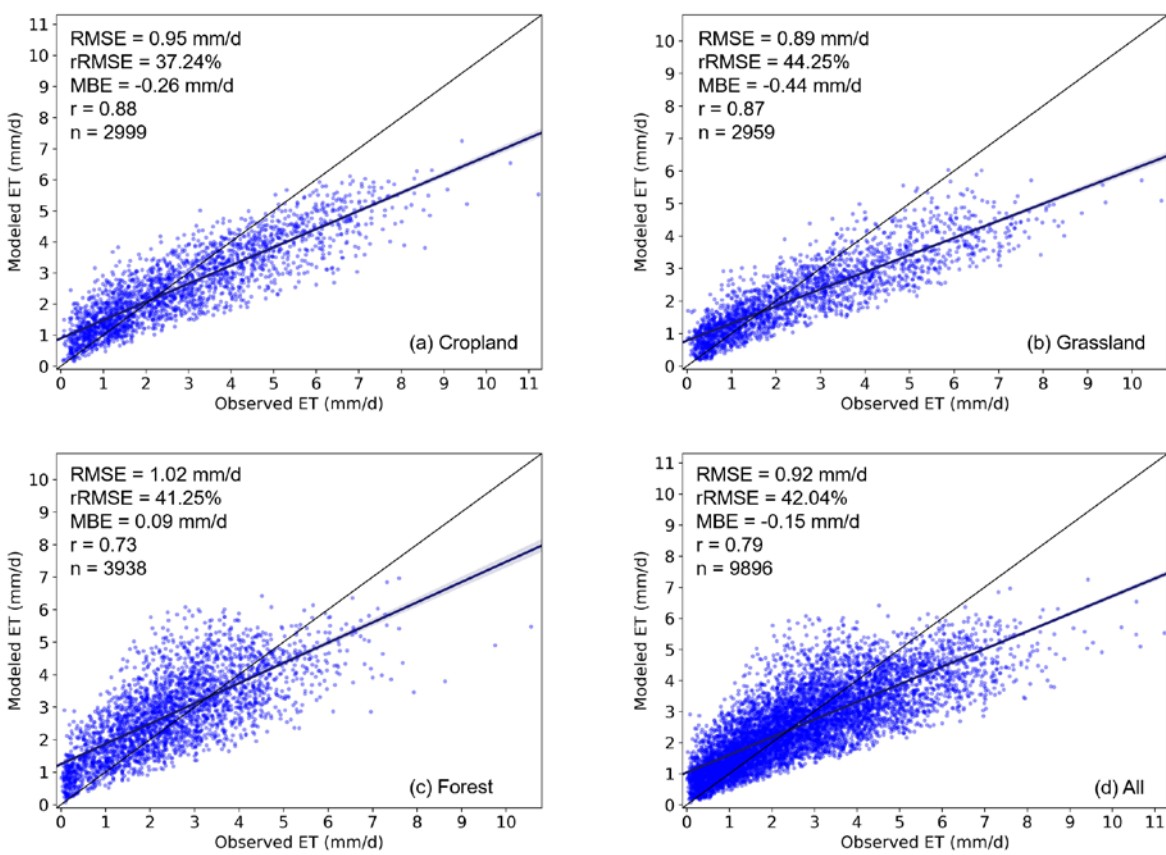

**Figure 3.** The validation of daily ET estimates using the SEBAL model and multisource images. (a) cropland; (b) grassland; (c) forest; (d) all land cover types.

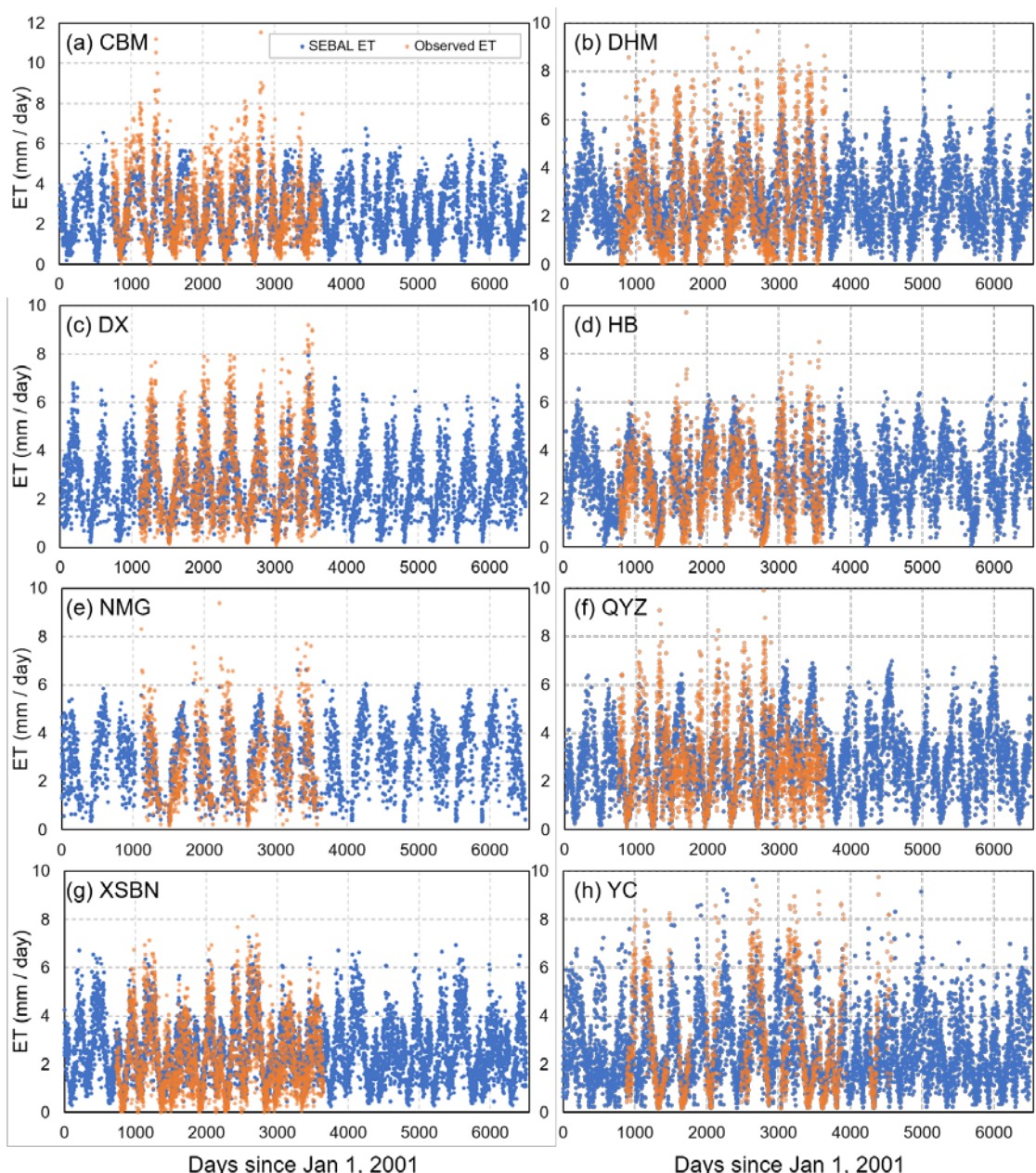

**Figure 4.** The SEBAL ET and flux tower observed ET variation in time series. (a) CBM; (b) DHM; (c) DX; (d) HB; (e) NMG; (f) QYZ; (g) XSBN and (h) YC.

**3.2 Comparison of SEBAL and MOD16 ET under different environmental conditions at the 8-day scale**

**3.2.1 Performance of the RS-based model for different land cover types**

The validation results for different land cover types are shown in Fig. 5. The results indicate that the accuracy of SEBAL and

MOD16 both varied with land cover type. The RMSE of SEBAL varied from 6.51 to 8.57 mm/8 d, its rRMSE varied from 38.08 to 52.63%, and its r-value varied from 0.81 to 0.87. The performance of SEBAL was superior for forest (RMSE = 8.54 mm/8 d, rRMSE = 38.08%) compared to other land cover types, and the lowest accuracy was obtained over grassland (RMSE = 6.51 mm/8 d, rRMSE = 52.63%). The results of the MOD16 validation indicate that MOD16 had a better performance for forest (RMSE = 8.88 mm/8 d, rRMSE = 39.29%) than other land cover types, as was observed for SEBAL, and the performance of MOD16 over grassland was also the worst (RMSE = 7.77 mm/8 d, rRMSE = 62.89%). The MBE values for MOD16 varied from 0.42 to 3.44 mm/8 d, which indicated that both the ET models underestimated ET over all land cover types. Overall, the accuracy of SEBAL was higher than that of MOD16.

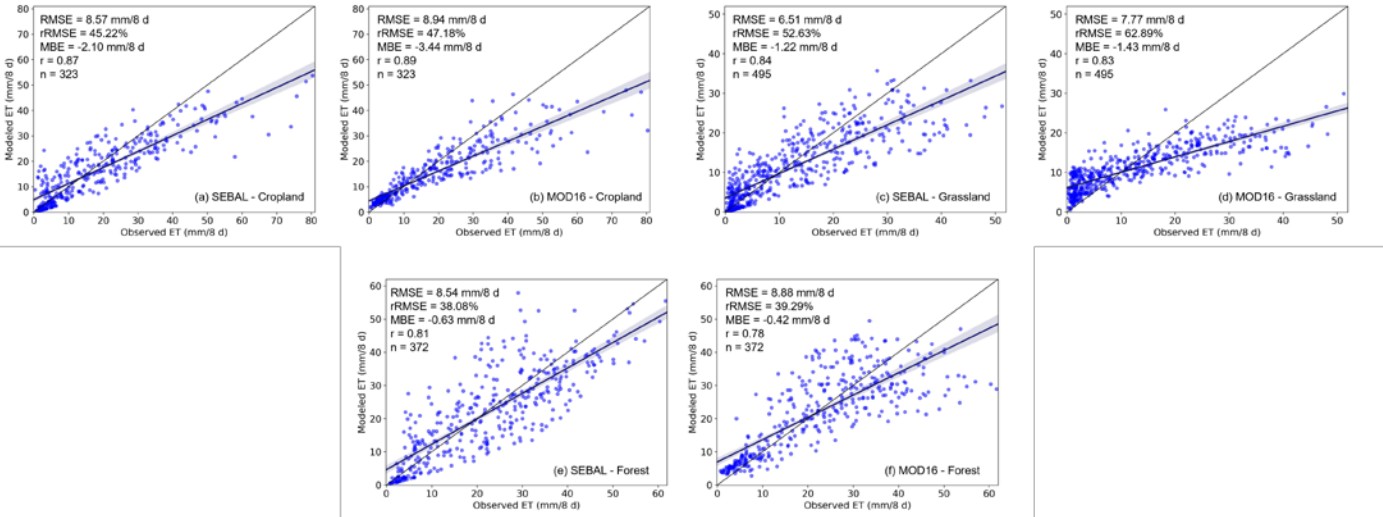

**Figure 5.** Validations for different land cover types. (a) SEBAL ET for cropland; (b) MOD16 ET for cropland; (c) SEBAL ET for grassland; (d) MOD16 ET for grassland; (e) SEBAL ET for forest; (f) MOD16 ET for forest.

**3.2.2 Performance of the RS-based model for different climate zones**

The validation results for different climate zones are shown in Fig. 6. The results show that the r-value varied from 0.68 to 0.90 for SEBAL and varied from 0.61 to 0.94 for MOD16. Climate zones were found to influence the accuracy of the RS-based models. In tropical zones, both the two models showed poor accuracy, with RMSEs of 10.75 and 11.37 mm/8 d for SEBAL and MOD16, respectively, and low r-values of 0.68 and 0.61 for SEBAL and MOD16, respectively. Additionally, both the models overestimated, with MBEs of 7.58 and 8.86 mm/8 d for SEBAL and MOD16, respectively. For subtropical zones, both the models had high precision, with rRMSEs of 32.32% and 36.73% for SEBAL and MOD16, respectively, and both underestimated, with r-values of 0.86 and 0.82 for SEBAL and MOD16, respectively. For warm temperate zones, both SEBAL and MOD16 showed poor accuracy, with rRMSEs of 53.95% and 56.12%, respectively, and both underestimated. For temperate zones, MOD16 overestimated, while SEBAL underestimated, and both models had high r-values, namely 0.90 for SEBAL and 0.94 for MOD16, and low RMSEs of 5.72

mm/8 d for SEBAL and 4.61 mm/8 d for MOD16. In general, MOD16 performed better than SEBAL for temperate zones. For alpine zones with low temperature, both the models still underestimated, however, SEBAL performed better than MOD16: the RMSE was 7.53 and 9.20 mm/8 d and the r-value was 0.79 and 0.77 for SEBAL and MOD16, respectively.

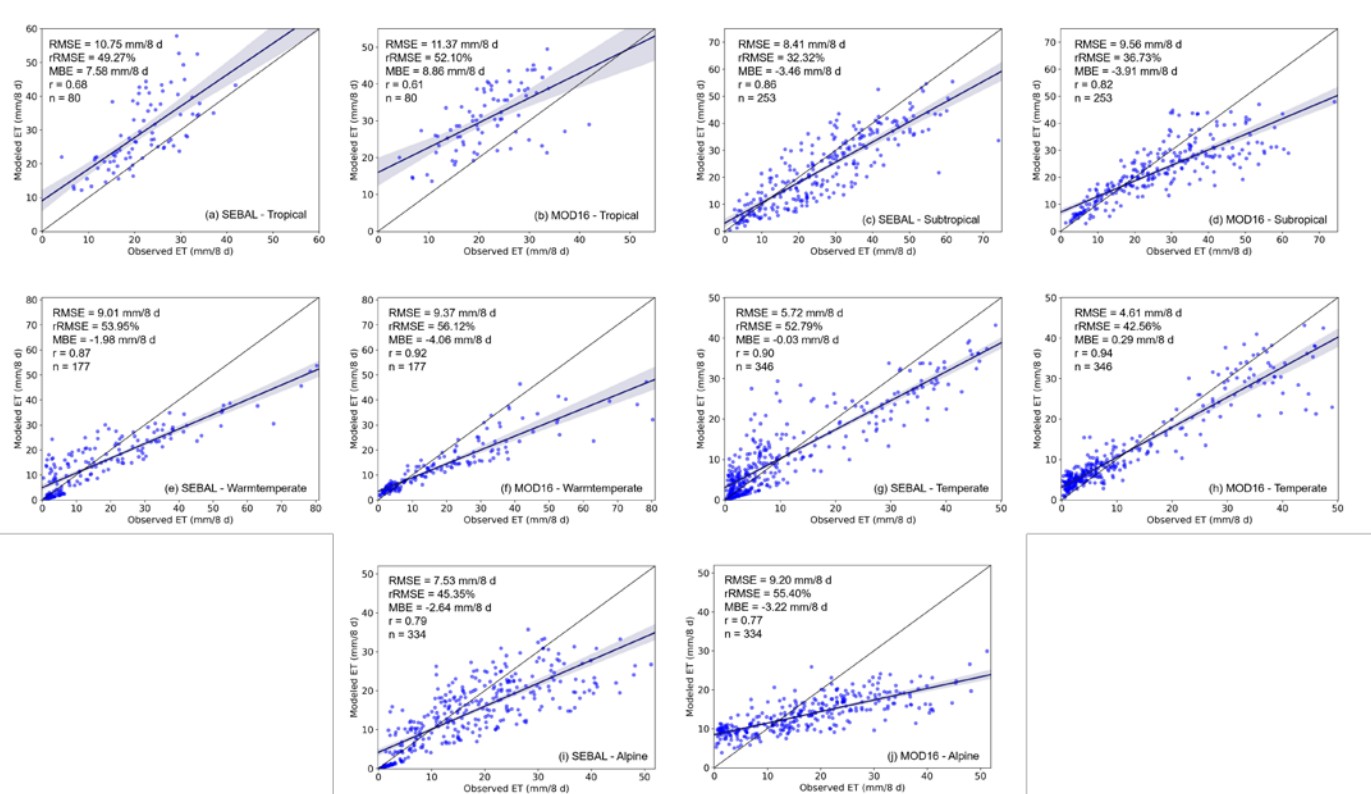

**Figure 6.** Validations for different climate zones. (a) SEBAL ET for tropical zones; (b) MOD16 ET for tropical zones; (c) SEBAL ET for subtropical zones; (d) MOD16 ET for subtropical zones; (e) SEBAL ET for warm temperate zones; (f) MOD16 ET for warm temperate zones; (g) SEBAL ET for temperate zones; (h) MOD16 ET for temperate zones; (i) SEBAL ET for alpine zones; (j) MOD16 ET for alpine zones.

**3.2.3 Performance of the RS-based model over different terrain types**

The validation results for different terrain types are shown in Fig. 7. The results indicate that both models showed a negative bias (negative MBE) for all terrain types except mountainous areas, for which both models overestimated, with MBEs of 1.19 and 1.67 mm/8 d for SEBAL and MOD16, respectively. In general, for mountainous areas, MOD16 showed a higher accuracy (RMSE = 7.79 mm/8 d, rRMSE = 41.88%, r = 0.82) than SEBAL (RMSE = 8.37 mm/8 d, rRMSE = 45.06%, r = 0.79). However, for all other

terrain types, SEBAL showed a higher accuracy. With SEBAL, the RMSE decreased from 9.01 to 6.51 mm/8 d as elevation increased. For hilly areas, SBEAL showed the lowest rRMSE (32.32%) while MOD16 showed the highest rRMSE (36.73%). For plain areas, SEBAL showed a slightly higher accuracy (RMSE = 9.01 mm/8 d, rRMSE = 53.95%) than MOD16 (RMSE = 9.37 mm/8 d, rRMSE = 56.12%), while for plateau area, SEBAL (RMSE = 6.51 mm/8 d, rRMSE = 52.63%) was more accurate than MOD16 (RMSE =

7.77 mm/8 d, rRMSE = 62.89%).

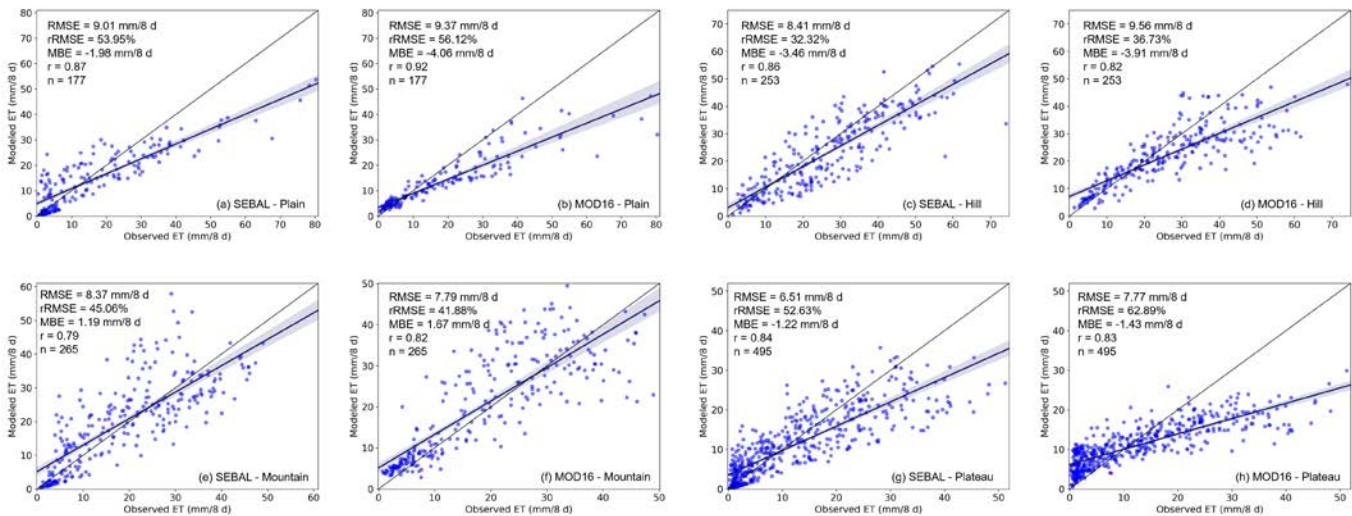

**Figure 7.** Validation over different terrain: (a) SEBAL ET in plain area; (b) MOD16 ET in plain area; (c) SEBAL ET in hill area; (d) MOD16 ET in hill area; (e) SEBAL ET in mountain area; (f) MOD16 ET in mountain area; (g) SEBAL ET in plateau area; (h) MOD16 ET in plateau area.

### 3.2.4 Performance of the RS-based model in different seasons

The validation results for different seasons are shown in Fig. 8. SEBAL showed a negative bias in summer, autumn, and winter, with MBE values varying from –2.95 to –0.62 mm/8 d, and showed a positive bias in spring (MBE = 3.13 mm/8 d). MOD16 showed a positive bias in winter (MBE = 3.8 mm/8 d) and a negative bias in other seasons, with MBE values varying from –0.58 to –0.50 mm/8 d. In spring, MOD16 generally showed a better performance (RMSE = 8.10 mm/8 d, rRMSE = 50.13% and r = 0.77) than SEBAL (RMSE = 9.18mm/8 d, rRMSE = 56.92% and r = 0.75), while SEBAL performed better than MOD16 in other seasons. In

winter, both the models showed a poor performance, with rRMSEs of 66.92% and 87.80% for SEBAL and MOD16, respectively. For both models, the highest accuracy was achieved in summer, with rRMSEs of 36.56% and 43.95% for SEBAL and MOD16, respectively. Meanwhile, the highest r-values were obtained in autumn, with values of 0.89 and 0.84 for SEBAL and MOD16, respectively.

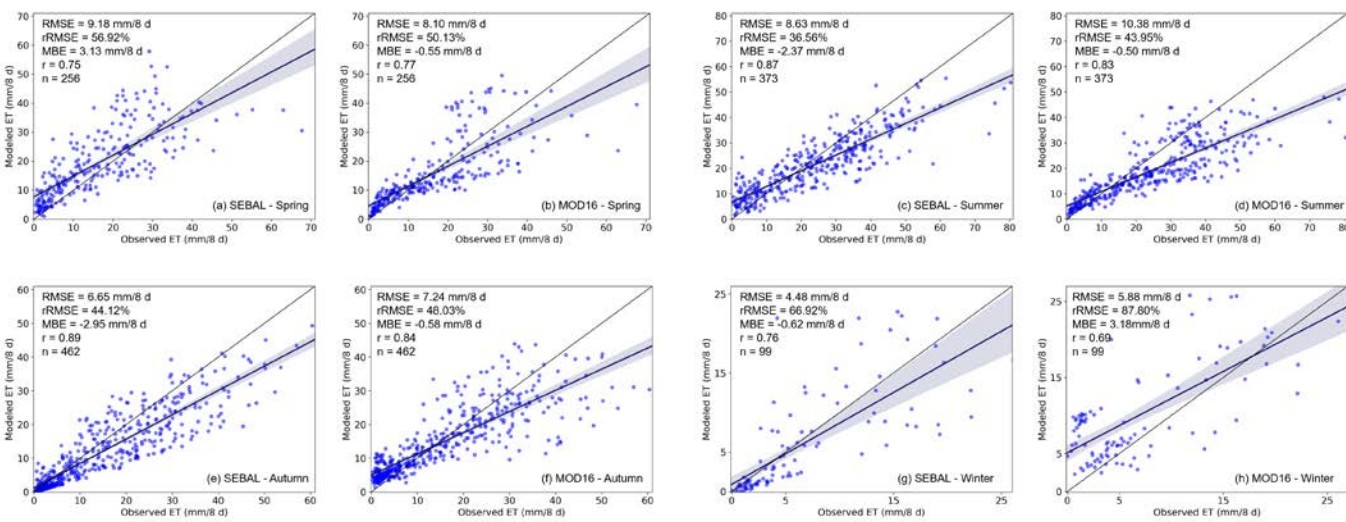

**Figure 8.** Validation for different seasons. (a) SEBAL ET for spring; (b) MOD16 ET for spring; (c) SEBAL ET for summer; (d) MOD16 ET for summer; (e) SEBAL ET for autumn; (f) MOD16 ET for autumn; (g) SEBAL ET for winter; (h) MOD16 ET for winter.

### 3.2.5 Summary of point-scale validation

Based on the contents of sections 3.1.1–3.1.4, SEBAL showed a higher accuracy than MOD16 in most conditions, while MOD16 showed a better performance only for temperate zones, mountainous areas, or the spring season based on the values of RMSE and rRMSE. Moreover, both the models underestimated for all conditions, except that SBEAL overestimated for tropical zones (MBE = 7.58 mm/8 d), mountainous areas (MBE = 1.19 mm/8 d), or spring (MBE = 3.13 mm/8 d), and MOD16 overestimated for tropical zones (MBE = 8.86 mm/8 d), temperate zones (MBE = 0.29 mm/8 d), mountainous areas (MBE = 1.67 mm/8 d), or winter (MBE = 3.18 mm/8 d). In general, SEBAL showed a higher accuracy than MOD16 based on point-scale validation (Fig. 9). For SEBAL and MOD16, respectively, the RMSE was 7.77 and 8.43 mm/8 d, the rRMSE was 44.91% and 48.72%, and the r-value was 0.85 and 0.83. Furthermore, both the models slightly underestimated overall, with an MBE of –1.27 and –1.66 mm/8 d for SEBAL and MOD16, respectively.

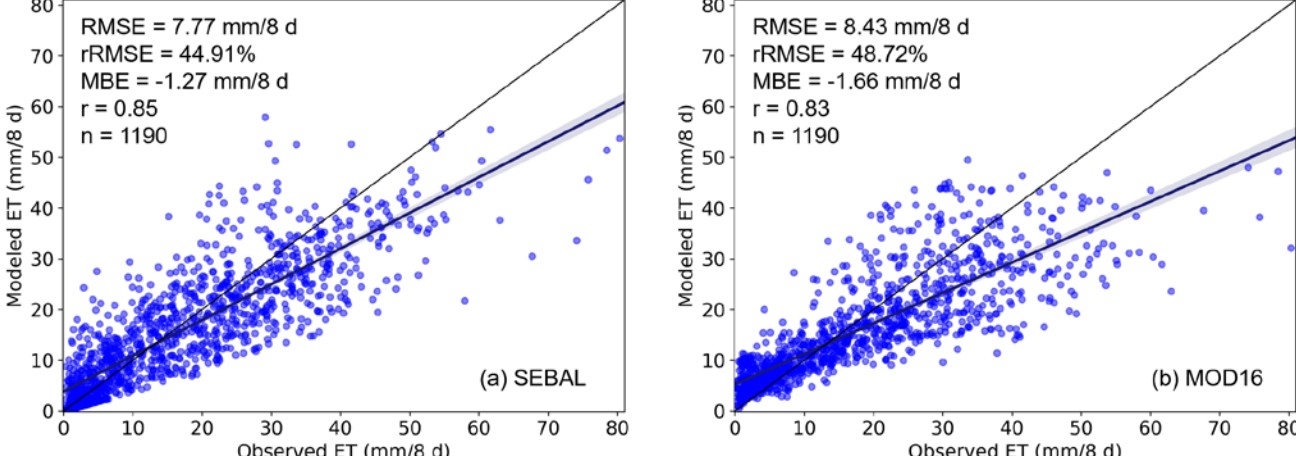

**Figure 9.** The results of the overall validation. (a) SEBAL ET validation at the 8-day scale; (b) MOD16 ET validation at the 8-day scale.

### 3.3 Validation at the basin-scale using the water balance method

Additionally, validation using hydrological data was performed to investigate the performance of the RS-based models at the basin-scale. The results (Fig. 10) showed that both the models had a negative bias, with an MBE of –24.45 and –96.66 mm/year for SEBAL and MOD16, respectively, at the basin-scale. SEBAL showed a higher accuracy, with an RMSE of 42.05 mm/year, an rRMSE of 12.65%, and an r-value of 0.98 (MOD16: RMSE = 118.55 mm/year, rRMSE = 32.84%, r = 0.91). As shown in Table 2, the average ET of SEBAL varied from 141.83 to 682.22 mm/year among the different basins, while bias varied from –95.86 to 39.67 mm/year. The average ET of MOD16 varied from 238.44 to 559.22 mm/year among the different basins, while bias varied from –281.04 to 98.43 mm/year.

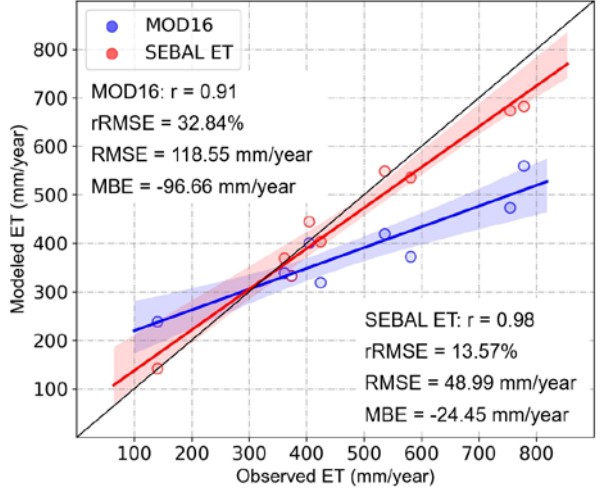

**Figure 10.** The results of validation at the basin-scale.

**Table 2.** The performance of the ET estimation of RS-based models at the basin-scale.

| Basin | Model | Average (mm/year) | Bias (mm/year) |
|---|---|---|---|
| SLRB | SEBAL | 369.11 | 8.06 |
| | MOD16 | 339.09 | -21.96 |
| HRB | SEBAL | 403.46 | -20.78 |
| | MOD16 | 319.43 | -104.81 |
| HuRB | SEBAL | 535.56 | -45.38 |
| | MOD16 | 372.45 | -208.49 |
| YeRB | SEBAL | 332.92 | -40.67 |
| | MOD16 | 333.18 | -40.41 |
| YRB | SEBAL | 549.04 | 13.34 |
| | MOD16 | 419.77 | -115.93 |
| PRB | SEBAL | 673.91 | -80.26 |
| | MOD16 | 473.13 | -281.04 |
| SeB | SEBAL | 682.22 | -95.86 |
| | MOD16 | 559.22 | -218.86 |
| SwB | SEBAL | 444.27 | 39.67 |
| | MOD16 | 400.52 | -4.08 |
| CB | SEBAL | 141.82 | 1.81 |
| | MOD16 | 238.44 | 98.43 |

Note: SwB: Southwest Basin; CB: Continental Basin; PRB: Pearl River Basin; YRB: Yangtze River Basin; SeB: Southeast Basin; HRB: Haihe River Basin; YeRB: Yellow River Basin; HuRB: Huaihe River Basin; SLRB: Songhua and Liaohe River Basin.

**3.4 Comparison of the spatial distribution of ET between SEBAL and MOD16**

Regarding the modeled spatial distribution of ET, both the SEBAL and MOD16 models showed that the annual average (2001–2018) ET in China increased from the northwest to the southeast (Fig. 11(a), (b)). Fig. 11(d). The annual ET of SEBAL varied from 0 to 1600 mm in space, with a mean value of 482.27±192.31 mm, while that of MOD16 varied from 0 to 1200 mm, with a mean value of 359.61±59.52 mm. In general, compared to the ET value estimated using MOD16 and SEBAL, the ET value estimated using SEBAL was higher and showed a greater spatial difference of ET in China. For 84.07% of the total area of China, the annual

ET estimated by SEBAL was higher than that estimated by MOD16; for 14.07% of the total area of China, the difference was more than two times—these areas are mainly distributed in Southern China, where ET is relatively high, and the difference reaches more than 600 mm in some places. Only in 15.93% of the total area of the country was the annual ET estimated by SEBAL lower than that estimated by MOD16; these areas are mainly distributed in Northwest China, where ET is relatively low (Fig. 11(c), (e)). Regarding the distribution of $ET_{SEBAL}$, a bi-modal curve with the boundary of ~500 mm was shown in the Fig. 11d, it was likely

contributed by the misestimation of part of regions. The ET$_{SEBAL}$ map was divided to two parts with 500 mm as threshold value, the part of ET$_{SEBAL}$ below 500 mm was distributed in the Northwest China (Fig. 11f), whereas the part of ET$_{SEBAL}$ over 500 mm was distributed in the southeast (Fig. 11g). It should be noted that the vegetation cover in northwest of China are mainly grassland and a small part of cropland (Fig. 11h), and the ET$_{SEBAL}$ of grassland and cropland was underestimated by SEBAL model (Section 3.1). In contrast, the ET$_{SEBAL}$ showed slightly overestimation of forest which is the main land cover types in southeast of China. Therefore,

the distributed ET$_{SEBAL}$ around ~500 mm was underestimated or overestimated, and thus formed the bi-modal curve.

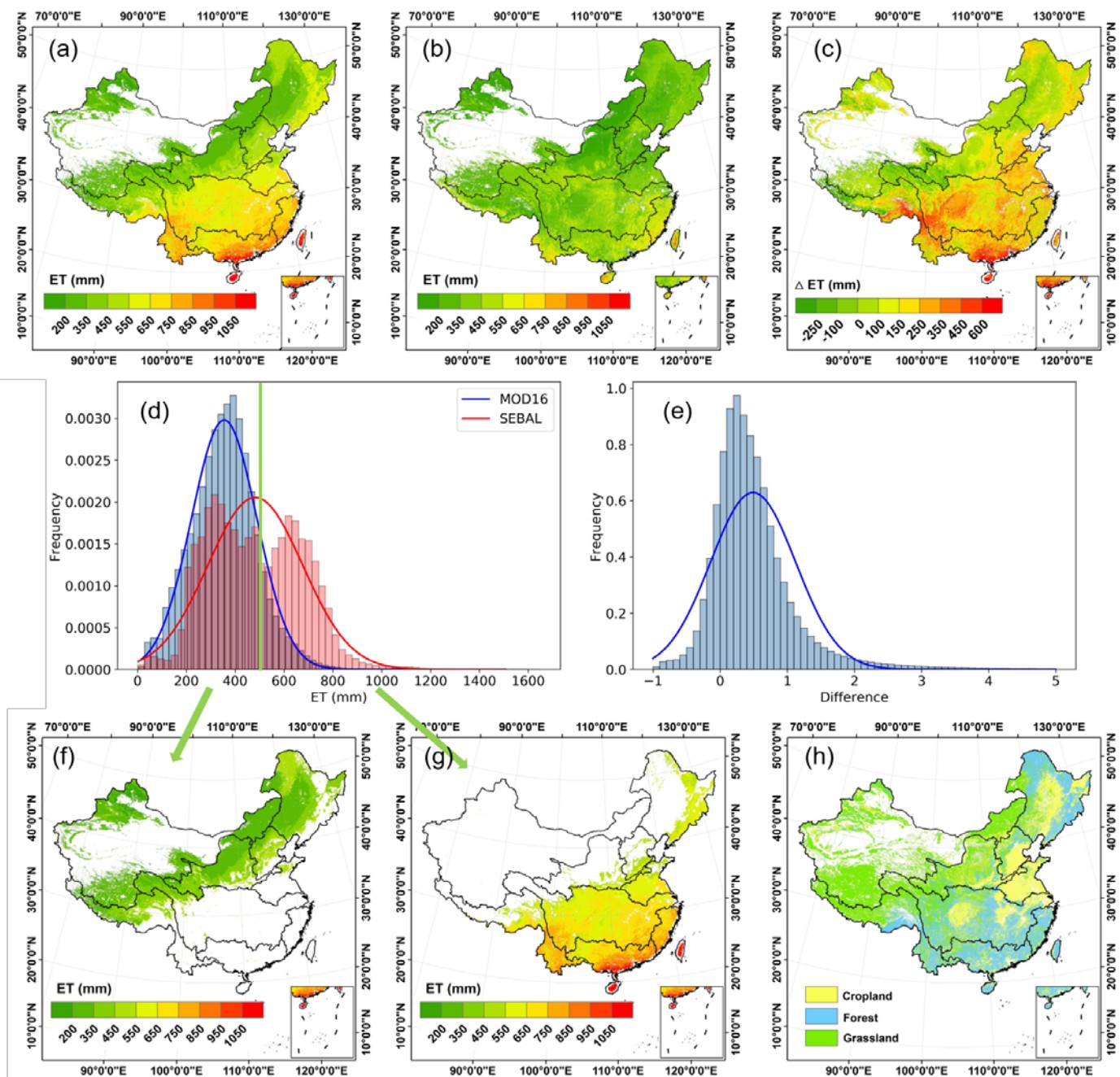

**Figure 11.** A comparison between the SEBAL and MOD16 models. (a) Distribution of annual average $ET_{SEBAL}$; (b) Distribution of annual average $ET_{MOD}$; (c) Distribution of the difference between SEBAL and MOD16 (ETSEBAL – ETMOD); (d) Histogram of annual average $ET_{SEBAL}$ and $ET_{MOD}$; (e) Histogram of the relative difference between SEBAL and MOD16 (($ET_{SEBAL}$ – $ET_{MOD}$)/$ET_{MOD}$); (f) Map of $ET_{SEBAL}$ below 500 mm; (g) Map of $ET_{SEBAL}$ over 500 mm; (h) Land cover in China.

## 4. Discussion

### 4.1 Summary of validation results and comparison with other studies

The $ET_{SEBAL}$ showed a relatively good performance in China as a whole, with an average r-value of 0.79 and an average RMSE of 0.92 mm/d. These results are close to those obtained in other studies. Rahimzadegan and Janani (2019) used SEBAL to estimate the actual ET of pistachio in Semnan, Iran, and found that the modeled value had a high consistency with the in-situ measured value (r = 0.80); this value was slightly lower than the cropland validation obtained in the present study (r = 0.88, daily-scale). This difference is mainly due to differences in the validation method between these two studies. Rahimzadegan and Janani (2019) used the P-M equation and field observational data from Intelligent Meteorological instruments to measure the standard ET, and MOD16 data which is also based on P-M equation was evaluated in this study that performed worse than SEBAL estimation at both point-scale and basin-scale. Xue et al. (2020) used pySEBAL (SEBAL in the Python environment) to estimate the ET of almonds, tomatoes, and maize in the Central Valley of California, USA, and showed that the r-value and RMSE of pySEBAL varied from 0.60 to 0.86 and from 1.08 to 1.79 mm/d, respectively; the authors used Landsat 8 OLI/TIRS images with a spatial resolution of 30 m × 30 m as the model input, which leads to a lower influence of mixed pixels compared to MODIS data with a spatial resolution of 1 km × 1 km. Wagle et al. (2017) evaluated the performance of SEBAL for ET estimation for sorghum based on flux tower observational data from Oklahoma, USA; the results showed that the r-value varied from 0.73 to 0.87 while the RMSE varied from 0.83 to 1.24 mm/d, which is basically in agreement with the results of this study (r-value varied from 0.68 to 0.90 and RMSE varied from 4.48 to 10.75 mm/8 d under different environmental conditions). MOD16 performed worse than SEBAL. Part of the bias is caused by objective factors such as the inaccuracy of the input data and the limitations of the validation methods. Meanwhile, other bias is contributed by the subjective factor of the inborn defects of the algorithms. These factors will be discussed in detail in Sections 4.2 and 4.3. Overall, the SEBAL ET showed an acceptable performance in China by comparing pervious studies.

### 4.2 Errors caused by objective factors

#### 4.2.1 Inaccuracy of input data

Both SEBAL and MOD16 used MODIS data as the main input images (e.g., MCD43 surface albedo, MOD13 NDVI, MOD11 surface temperature). However, the accuracy of these data is uncertain to some extent (Ramoelo et al., 2014). For instance, surface

albedo is a critical radiative parameter, however, the complex algorithm-led remote sensing-based albedo products can contain errors introduced by the spectral conversion (Song et al., 2020). Wang et al. (2014) compared MODIS albedo products with ground data and Landsat data for different land cover types in the USA, and found that the RMSE of the products varied from 0.01–0.05 and that the error was higher during periods of snow cover. Furthermore, surface temperature, as a fundamental parameter for the calculation of surface energy balance, affected the estimation of ET to a great extent (Long et al., 2011). Timmermans et al. (2006) analyzed the sensitivity of each parameter of the SEBAL model to grassland in Oklahoma, USA, and the results indicated that the difference in surface air temperature had the greatest influence on the accuracy of SEBAL estimation. MODIS surface temperature products are retrieved using the split-window algorithm. Yu et al. (2019) used in-situ measurement data to validate MODIS surface temperature products in the Heihe River Basin (HRB) in Northwest China; the results indicated that the daytime MOD11 (obtained by the Terra satellite) and MYD11 (obtained by the Aqua satellite) products have accuracies of –0.84±0.88 K and –0.11±0.42 K, respectively. In general, original MODIS data performed errors to some extent.

Additionally, gap-filling of missing or unreliable MODIS data may causes the errors to some extent. For example, spring and summer have the relatively frequent precipitation, which causes more unreliable pixels due to the cloud, and these pixels value were finally replaced by gap-filling of nearest date pixel value, therefore, the modeled ET value of these pixels was close to that of nearest date without precipitation. In fact, due to the high air humidity in rainy day, the evaporation and transpiration are relatively less than that of nearest date (Ferreira and Cunha, 2020; Li et al., 2016). Moreover, it should be noted, due to the decrease of surface available radiation energy which was caused by cloud cover, the ET (both actual and modeled value) is also less than that of nearest date (Cheng et al., 2020). This may explain the reason of obvious overestimation at lower ET rates in spring, summer and other pixels affected by cloud. Furthermore, a relatively high bias of SEBAL ET was found in winter, the rRMSE reached 66.92% (the highest value among all situations). Due to the ice and snow cover caused by the frequent snowfall and low temperature in winter, which will affect the remotely sensed information to a great extent, e.g., reflectance (Casey et al., 2017), and further affect the ET estimation. Moreover, the underestimation was found at higher ET rates in the most of situations as shown in Figs. 4-10, which may cause by the saturation issue of optical sensor (Maimaitijiang et al., 2020). For example, in the dense vegetation covers, the vegetation index (e.g., NDVI) was likely underestimated and can not accurately characterize vegetation status (Maimaitijiang et al., 2020; Omar et al., 2016), therefore, soil heat flux will be overestimated (Eq. 9 in appendix), and may further caused the latent heat flux underestimation.

### 4.2.2 Errors in flux tower measurements

The eddy covariance system (flux tower observations) is the most commonly used observation system to calculate and analyze the energy and mass exchange between the surface and atmosphere (Wang and Dickinson, 2012). However, the typical error of ET

estimation based on the eddy covariance system is about 5–20% (Culf et al., 2008; Vickers et al., 2010). Also, the eddy covariance system generally has an energy balance non-closure issue that, the sum of the soil heat flux, sensible heat flux and latent heat flux was found to be less than net radiation in most cases (Mu et al., 2011; Wilson et al., 2002). Recently, it was found that the non-closure issue of the energy balance was explained by the energy fluxes from secondary circulations and larger eddies that cannot be

captured by EC measurement at a single station (Foken et al., 2011). In this study, the Bowen ratio method (Eq. 3), which assuming that the residual of the energy balance is attributed to sensible and latent heat flux and assigning the missing energy flux to them (Song et al., 2016; Wang et al., 2019), was used to enforce energy closure. Actually, this assumption is not very correct, which generally led the sensible and latent heat flux overestimation (Song et al., 2016), which may could explain that the SEBAL ET was generally underestimated when compared to flux tower observed ET (Fig. 9). The same issue was found in regional-scale validation,

due to the ignoring of $\Delta S$ in the water balance computation process (although it's small), which could lead the regional ET overestimation and further caused SEBAL ET underestimation in regional validation (Fig. 10).

Additionally, the mismatch of flux tower footprint and spatial resolution of SEBAL ET will causes errors as well. Generally, the footprint of flux tower varied from hundreds square meters to several square kilometers which determined by the height of the observation instrument, the intensity of the turbulence, terrain, environment and vegetation status (Chen et al., 2012; Damm et al.,

2020; Schmid, 1994). Moreover, a footprint probability distribution function (PDF) could characterize the footprint in a fine spatial resolution (Wang et al., 2019), but it may not suit for the coarse resolution in this study (kilometer-scale). In this study, the 1 km × 1 km area of pixel was used for matching the footprint of flux tower which was referred to the study of Velpuri et al. (2013), however, the footprint is not stable but varied with environment changed, e.g., vegetation height. Chen et al. (2012) reported that forest footprint has clear difference with grassland, the footprint of forest is much larger which reach kilometer-scale. In fact, forest

footprint may more matching with the spatial resolution in this study. Therefore, it may explain that the SEBAL ET has the greatest performance in forest but worst performance in grassland. Compared to the study of Velpuri et al. (2013), the grassland also showed the worst remote sensing ET estimation in US when using flux tower data for validation at a kilometer-scale.

Although a comprehensive evaluation of SEBAL ET over different classes was conducted in this study, users should be aware of the uncertainties due to the limited number of validation sites in some classes. For example, only one site was available for the

435 evaluation over cropland. Because this cropland flux tower site was set in plain and warm-temperate zone, the accuracy may only represent the data quality of cropland ET in the warm-temperate plain zone, but not other regions. Nevertheless, long-time-series data were obtained from this site which covered different seasons and different crop types. Employing these hundreds of samples in the validation could remedy the single-site insufficiency to a certain extent. Similarly, only one site was found in the validation over two other classes, i.e., tropical zone and warm-temperate zone. Long-time-series data were also incorporated to enhance the

440 representativeness of the single site. Regarding the other classes, two or more sites were used which will lead to more reliable results. Compared to pervious studies (Aguilar et al., 2018; Hu et al., 2015; Ramoelo et al., 2014; Yang et al., 2017), a larger number of

validation samples (flux tower sites) were used in this study, indicating that the findings were reliable. Additionally, although the validation of SEBAL ET in this study followed the literature (Kim et al., 2012; Ramoelo et al., 2014). and considered different land cover types, climate zones, elevation and seasons, several more situations may need to be considered. For example, whether the SEBAL accuracy was different across years (Velpuri et al., 2013) and satellite sensors (Long et al., 2011). Overall, with the increasing number of flux towers set up in China, more reliable and comprehensive validation of SEBAL ET can be conducted in the follow-up research.

## 4.3 Errors caused by subjective factors

### 4.3.1 Temporal scaling-up method

Remotely sensed information represents the information of satellite-passing time. Therefore, in the RS-based models, scaling-up was performed from the instantaneous level to the daily level. SEBAL uses the evaporative fraction ($\Lambda$) for scaling-up (Gao et al., 2020), as shown in Eqs. 32 and 33. However, several studies have indicated that the assumption of a constant evaporative fraction is not very reasonable (Gentine et al., 2011; Hoedjes et al., 2008). Gentine et al. (2007) proposed that soil moisture and vegetation resistance are the factors that mainly affect the stability of $\Lambda$, and soil moisture is positively correlated with $\Lambda$. Additionally, a larger leaf area index will generally lead to a lower stability of $\Lambda$ under the same soil moisture (Farah et al., 2004). In general, due to the instability of $\Lambda$, the above assumption will cause a negative bias of 10–20% in the estimation of daily ET (Delogu et al., 2012; Ryu et al., 2012; Van Niel et al., 2012). This can explain why the validation in this paper showed that the ET estimated using SEBAL was underestimated. While the MOD16 model estimates daily ET using the P-M equation, which is a semi-empirical equation, it uses 8-day or 16-day composite remotely sensed input data and daily meteorological input data to compute the 8-day composite ET products (Mu et al., 2011). The use of a semi-empirical equation avoids the need to perform scaling-up, however, it has the problem of theoretical deficiency (Mu et al., 2007; Ramoelo et al., 2014).

### 4.3.2 Calculation of sensible heat flux

Sensible heat flux is the most complicated part of the energy balance calculation (Wang and Dickinson, 2012). The P-M algorithm defines the available energy (A, unit: W/m$^2$) as the sum of the sensible heat flux and latent heat flux (Eq. 10) (Mu et al., 2011).

$$A = H + \lambda ET = R_n - G \tag{10}$$

The P-M algorithm calculates $\lambda$ET using a semi-empirical formula and A, and therefore avoids the direct calculation of H. Meanwhile, SEBAL calculates H based on MOST and the hot/cold pixel (Bastiaanssen et al., 1998a; Bastiaanssen et al., 1998b). However, several studies have indicated that MOST has an error of 10–20% for the estimation of the boundary layer thickness (Foken, 2006; Högström and Bergström, 1996). Therefore, MOST is also a source of error in SEBAL. Due to the complexity of the

sensible heat flux, SEBAL makes several assumptions to estimate H, which may introduce error into the ET estimation (Zheng et al., 2016).

Additionally, the selection of the hot/cold pixel depends on the domain size (the actual size of the modeling domain/satellite imagery being used). For instance, the basin-scale selection of the hot/cold pixel with diverse vegetation cover and single vegetation cover,
respectively, will lead to different results for dT. Theoretically, with the domain size increasing, there is a possible tendency that $T_{s\_hot}$ increasing and $T_{s\_cold}$ decreasing. For example, if $T_{s\_cold}$ remains invariant and $T_{s\_hot}$ increases under the condition of domain size increasing, the H estimates will decrease and λET estimates could thus increase. In the study of Long et al. (2011), the results showed that a 2 K increase in $T_{s\_hot}$ will result in a 9.3% increase but a 9.1% decrease in *a* and *b*, respectively, and further caused an 11.8% mean decrease in H. Recently, the study of Saboori et al. (2021) reported that the cold pixel performed more stable than
hot pixel in time series, especially in winter, the hot pixel was highly varied may due to the similarity of NDVI over space, it could further explain the poor performance of SEBAL ET in winter. Seguin et al. (1999) demonstrated the method of hot/cold pixel selection for the estimation of H generally has an accuracy of ~50 W/m². In this study, the ET was computed in the domain size of 1200 km × 1200 km which is a relatively large area. The performance of different domain sizes in ET computation were compared (Fig. 12) and resulted in better performance are generally found in smaller area, which may due to the relatively limited spatial
heterogeneity (Long et al., 2011). However, a larger domain size may have faster computational efficiency of ET in regional scale. The trade-off between efficiency and accuracy (i.e., most suitable domain size) need be further studied. Overall, the domain size employed in this study (1200 km × 1200 km) performed an acceptable accuracy. Although several algorithms have been proposed that use other methods to avoid the error caused by the selection of the hot/cold pixel , such as the SEBS (Su, 1999), these replaced the selection of the hot/cold pixel with the fitting of dry and wet edges. However, no evidence has been found that the method of
fitting dry and wet edges can significantly improve the accuracy of ET estimation (Wagle et al., 2017; Xue et al., 2020).

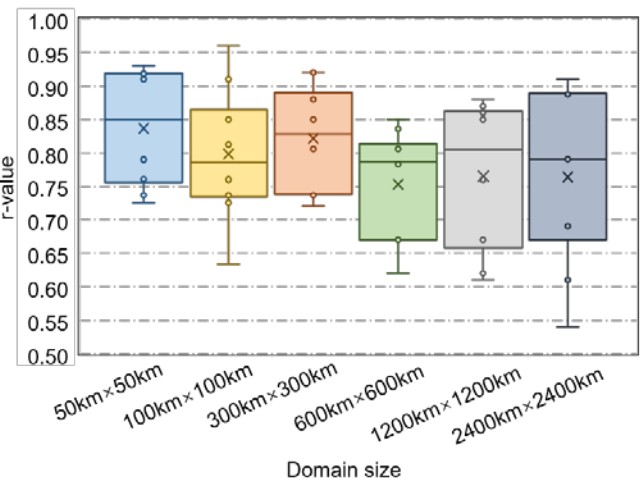

**Figure 12.** The performance of different domain sizes used in SEBAL ET estimates.

Besides sensible heat flux, the errors of SEBAL ET may derived from net radiation or soil heat flux as well (Li et al., 2017; Teixeira et al., 2009). For net radiation, which is computed using surface albedo and Stephen Boltzmann law (Eq. 2 in appendix), generally

performed a relatively agreement with flux tower observed value, while soil heat flux, which computed using empirical formula related to net radiation and NDVI (Eq. 9 in appendix), has a poor performance (Li et al., 2017; Song et al., 2016). In the study of Li et al. (2017), soil heat flux estimation showed a clear overestimation in higher ET area, e.g., wetland, which may further caused the sensible and latent heat flux underestimation in higher ET rates. In the most SEBR-based algorithms, the similar net radiation and soil heat flux estimation methods are used, and various sensible heat flux estimation methods are the main sources of the difference among the various SEBR-based algorithms. However, the causes of the net radiation and soil heat flux estimation errors have not been clearly discussed, e.g., the effect of satellite transmitted time or land cover types. These issues could be the focus of our follow-up research, for example, geostationary satellite and flux tower with high frequency observations may be helpful for this research.

## 5. Data availability

The dataset that was generated using SEBAL with a spatial resolution of 1 km and a temporal resolution of 1 day can be used for various types of geoscientific studies, especially for global change, water resources management, agricultural drought monitoring, etc. The evapotranspiration (ET) dataset for China is distributed under a Creative Commons Attribution 4.0 International license. The dataset is named SEBAL evapotranspiration in China (SEBAL ET) and consists of 18 years of data. More information and data are freely available from the Zenodo repository at https://doi.org/ 10.5281/zenodo.4243988 (Cheng, 2020).

## 6. Conclusions

In this study, we generated a long time series (2001–2018) ET product based on SEBAL and multisource images. We further conducted a comprehensive validation of the product and compared its performance under different environmental conditions in China with the performance of the ET estimated using MOD16 data. The conclusions are as follows:

(1) The ET product generated using SEBAL showed a good performance in China. Compared to flux tower observational data, the r-value of the SEBAL ET reached 0.79 for 9896 samples; the RMSE was 0.92 mm/d and the rRMSE was 42.04%. SEBAL underestimated ET as whole, with an MBE of –0.15 mm/d. The SEBAL ET product can adequately represent the actual ET and can be used in research on water resources management, drought monitoring, ecological change, etc.

(2) Based on observational data from eight flux towers from 2003 to 2010, the ET datasets estimated using SEBAL and MOD16 were validated at the 8-day scale for different land cover types, climate zones, terrain types, and seasons. The results showed that SEBAL performed best in the conditions of forest cover (rRMSE = 38.08%), subtropical zones (rRMSE = 32.32%), hilly terrain (rRMSE = 32.32%), and the summer season (rRMSE = 36.56%), respectively, and performed worst in the conditions of grassland cover (rRMSE = 52.63%), warm-temperate zones (rRMSE = 53.95%), plain terrain (rRMSE = 53.95%), and the winter season (rRMSE = 66.92%), respectively; MOD16 performed best in the conditions of forest cover (rRMSE = 39.29%), subtropical zones

(rRMSE = 36.73%), hilly terrain (rRMSE = 36.73%), and the summer season (rRMSE = 43.95%), respectively, and performed worst in the conditions of grassland cover (rRMSE = 62.89%), warm-temperate zones (rRMSE = 52.10%), plateau terrain (rRMSE = 62.89%), and the winter season (rRMSE = 87.80%), respectively. In general, the two models have similar adaptability to different conditions, although SEBAL performed slightly better than MOD16.

(3) Based on flux tower observational data and hydrological observational data, the ET estimated by SEBAL and MOD16 were validated at the point-scale and basin-scale. The results showed that, at the point-scale, the accuracy of SEBAL was 7.77 mm/8 d for the RMSE, 44.91% for the rRMSE, and 0.85 for the r-value, and the accuracy of MOD16 was 8.43 mm/8 d for the RMSE, 48.72% for the rRMSE, and 0.83 for the r-value. At the basin-scale, the accuracy of SEBAL was 48.99 mm/year for the RMSE, 13.57% for the rRMSE, and 0.98 for the r-value. In general, SEBAL performed slightly better than MOD16 at the point-scale, while SEABAL

had a larger accuracy advantage at the basin-scale.

(4) Overall, the SEBAL ET is higher than the MOD16 ET: for 84.07% of the total area of China, the SEBAL ET showed higher values. Additionally, the SEBAL ET is closer to the in-situ measured ET in most conditions, while the MOD16 ET performed better only in temperate zones, mountain areas, or the spring season. In general, the two models both have a good performance and can be

used in the qualitative analysis and most quantitative analysis of regional ET. Furthermore, the combination of the two models can improve the overall ET estimation accuracy for use in applications with higher accuracy requirements.

Compared to the widely used MOD16 ET data, the SEBAL ET product showed a higher accuracy and temporal resolution. However, it still has a daily error of 42.04% (0.92 mm/d) at the point-scale and a yearly error of 13.57% (48.99 mm/year) at the basin-scale.

Therefore, the improvement of the SEBAL algorithm will be the focus of follow-up research. Moreover, the 1 km spatial resolution of the SEBAL ET product cannot meet the requirements of more detailed research. Due to the difficulty of simultaneously satisfying the requirements for the spatial and temporal resolutions of remote sensing data, the fusion of multiple sources of remote sensing data may be the most effective way to improve the spatiotemporal resolution of daily ET products.

**Acknowledgements:**

This study was financially supported by the National Key Research and Development Program of China (2016YFD0300605), National Natural Science Foundation of China (grant 42071426), and Central Public-interest Scientific Institution Basal Research Fund for Chinese Academy of Agricultural Science (grant Y2020YJ07).

**Conflict of interest statement:**

The authors declare no conflict of interest.

**Appendix 1:Description of generating SEBAL ET in details**

The SEBAL model calculates the instantaneous λET of the satellite transit time as a residual based on the surface energy balance

equation (Eq. 1) as follows:

$$\lambda ET = R_n - G - H \tag{1}$$

where $R_n$ is the net radiation flux, $H$ is the sensible heat flux, and $G$ is the soil heat flux (the unit of all three parameters is W/m$^2$).

In this paper, MODIS data (MCD43 surface albedo, MOD11 surface temperature, MOD13 NDVI) and meteorological data (air

temperature) from the Global Modeling and Assimilation Office (GMAO) were used as input for surface parameterization ($R_n$, G

and H). The equations for $R_n$ are shown in Eqs. 2–5 below:

$$R_n = (1-\alpha)R_{s\_down} + R_{l\_down} - R_{l\_up} \tag{2}$$

where $\alpha$ is the surface albedo obtained from the MCD43 data; $R_{s\_down}$, $R_{l\_up}$, and $R_{l\_down}$ are the downwelling shortwave radiation,

downwelling longwave radiation, and upwelling longwave radiation, respectively (the unit of all three parameters is W/m$^2$). $R_{s\_down}$

can be calculated using the Julian day (used to estimate the astronomical distance between the sun and earth), elevation (used to

570 estimate atmospheric emissivity), and solar zenith angle at the time of satellite transit. $R_{l\_up}$ and $R_{l\_down}$ can be calculated using the

surface temperature (MOD11), NDVI (MOD13, used to estimate surface emissivity) and air temperature (GMAO data), and

atmospheric emissivity based on the Stefan-Boltzmann law. The equations for $R_{s\_down}$, $R_{l\_up}$, and $R_{l\_down}$ are given in Eqs. 3–5.

$$R_{s\_down} = \frac{G_{sc} \times \cos\theta \times \tau_{sw}}{d_r^2} \tag{3}$$

$$R_{l\_up} = \varepsilon_a \sigma T_a^4 \tag{4}$$

$$R_{l\_down} = \varepsilon \sigma T_s^4 \tag{5}$$

where $G_{sc}$ is the solar constant (1376 W/m$^2$); $\theta$ is the solar zenith angle; $\tau_{sw}$ is the atmospheric transmittance (Eq. 6) (Tasumi, 2000);

$d_r$ is the astronomical distance between the sun and earth (Eq. 7) (Bastiaanssen et al., 1998a); $\varepsilon_a$ and $\varepsilon$ are the atmospheric emissivity

(Eq. 8) (Bastiaanssen et al., 1998a) and surface emissivity (obtained from MOD11), respectively; $\sigma$ is the Stefan-Boltzmann constant

(5.67 × 10$^{-8}$ W/m$^2$K$^4$); and $T_a$ and $T_s$ are the air temperature (unit: K; obtained from GMAO data) and surface temperature (unit: K,

obtained from MOD11), respectively.

$$\tau_{sw} = 0.75 \times 2 \times 10^{-5} \times Z \tag{6}$$

$$dr = 1 + 0.0167\sin(\frac{2\pi(J-93.5)}{365}) \tag{7}$$

$$\varepsilon_a = 1.08(-\ln \tau_{sw})^{0.265} \tag{8}$$

where Z is the elevation obtained from a DEM (unit: m) and J is the Julian day. G can be calculated by the following empirical equation (Bastiaanssen et al., 1998a):

$$G = R_n \times \frac{T_s - 273.16}{\alpha}(0.0032 \times \frac{\alpha}{c} + 0.0032(\frac{\alpha}{c})^2) \times (1 - 0.978NDVI^4) \tag{9}$$

where $T_s$ is the surface temperature (unit: K) and $c$ represents the influence of the satellite transit time on $G$. The value of $c$ is 0.9 for transmission times before 12:00 local time (LT), 1.0 for transmission times between 12:00 and 14:00 LT, and 1.1 for transmission times between 14:00 and 16:00 LT. H can be calculated as follows:

$$H = \frac{\rho_{air}C_p dT}{r_a} \tag{10}$$

where $\rho_{air}$ (unit: kg/m$^3$) is the air density (Eq. 11) (Smith et al., 1991); $C_p$ (unit: J/(kg$\times$K)) is the specific heat of air at constant pressure; $dT$ (unit: K) is the difference between the aerodynamic surface temperature ($T_{z0h}$; unit: K) and the reference height temperature ($T_a$, unit: K); and $r_a$ is the aerodynamic resistance (unit: s/m) (Eq. 12).

$$\rho_{air} = 349.635 \frac{(T_a - 0.0065Z)^{5.26}}{T_a^{6.26}} \tag{11}$$

$$r_a = \frac{\ln(\frac{Z_2}{Z_1})}{kU_f} \tag{12}$$

where $k$ is the von Karman constant (0.41); $U_f$ is the frictional wind speed (unit: m/s) (Eq. 13); and $Z_1$ and $Z_2$ are 0.01 and 2, respectively.

$$U_f = \frac{kU_r}{\ln(Z_r / z_{om})} \tag{13}$$

where $U_r$ is the wind speed at height $Z_r$, which can be calculated from the wind speed monitored by weather stations ($U_w$, Eq. 14); $Z_r$ is 200 m in this study (Zeng et al., 2008); and $z_{0m}$ is the surface roughness (unit: m, Eq. 15) (Moran and Jackson, 1991).

$$U_r = \frac{U_w \times \ln(67.8Z_r - 5.42)}{4.87} \tag{14}$$

$$z_{0m} = e^{(5.65NDVI - 6.32)} \tag{15}$$

However, since it is difficult to calculate dT directly, the model assumes that there is a linear relationship between surface temperature ($T_s$, unit: K) and dT, as shown in Eq. 16:

$$dT = aT_s + b \tag{16}$$

SEBAL solves the values of $a$ and $b$ by selecting the hot and cold pixels; it assumes that the hot pixel represent pixel of dry cropland with low vegetation covers or bare surfaces or saline alkali land covered by vegetation with zero λET (H = $R_n$ - G), and the cold

pixel represent pixel with sufficient water supply, lush vegetation, and low temperature, with an H of zero ($\lambda ET = R_n - G$). In this study, the hot and cold pixels were selected automatedly by following the certain rules (Long et al., 2011): for hot pixel, the pixels with high Ts (top 10%) and low NDVI (top 10%) in the image were selected first, and further to select the pixels with the land covers of cropland or bare surfaces (according to MOD12 land use product) from the pixels selected in last step, finally, the pixel with highest Ts in these pixels was selected as the hot pixel. In contrast, for cold pixel, the pixels with low Ts (top 10%) and high NDVI (top 10%) in the image were selected first, and further to select the pixels with the land covers of dense vegetation (generally forest) from the pixels selected in last step, finally, the pixel with lowest Ts in these pixels was selected as the cold pixel. It should be noted that China area is made up by 28 tiles of remote sensing image (MODIS data), and each tile was computed independently, as well as hot and cold pixels selection independent in the ET generating process (Long et al., 2011). After hot and cold pixels determined, $a$ and $b$ can be expressed as follows:

$$a = \frac{(R_{n\_hot} - G_{hot})r_{a\_hot}}{C_p \rho_{air\_hot}(T_{s\_hot} - T_{s\_cold})} \tag{17}$$

$$b = -aT_{s\_cold} \tag{18}$$

Moreover, it should be noted that H and $r_a$ are interrelated variables in the actual calculation; therefore, the Monin–Obkhov Similarity Theory (MOST)-based Monin–Obkhov length (L, unit: m) is introduced for iterative calculation to obtain stable values of H and $r_a$. The details of MOST are shown in Fig. A1.

The Monin–Obkhov length is a parameter reflecting the turbulent characteristics of the near-surface layer (Eq. 19) (Monin and Obukhov, 1954); $\Psi_m(Z_r)$ is the stability correction function of momentum; and $\Psi_H(Z_1)$ and $\Psi_H(Z_2)$ are the stability correction functions of sensible heat flux (Eqs. 20–26) (Paulson, 1970).

$$L = \frac{\rho_{air}C_p U_f^3 T_s}{kgH} \tag{19}$$

where g is the acceleration due to gravity (9.81 m/s$^2$). While L>0, indicating a stable state, $\Psi_m(Z_r)$, $\Psi_H(Z_1)$, and $\Psi_H(Z_2)$ are calculated as follows:

$$\Psi_m(Z_r) = \frac{-5Z_r}{L} \tag{20}$$

$$\Psi_H(Z_1) = \frac{-5Z_1}{L} \tag{21}$$

$$\Psi_H(Z_2) = \frac{-5Z_2}{L} \tag{22}$$

While L<0, indicating an unstable state, $\Psi_m(Z_r)$, $\Psi_H(Z_1)$, and $\Psi_H(Z_2)$ are calculated as follows:

$$\Psi_m(Z_r) = 2\ln(\frac{1+\zeta_{Z_r}}{2}) + \ln(\frac{1+\zeta_{Z_r}^2}{2}) + 2\arctan(\zeta_{Z_r}) + 0.5\pi \tag{23}$$

$$\Psi_H(Z_1) = 2\ln(\frac{1+\zeta_{Z_1}^{\ 2}}{2}) \tag{24}$$

$$\Psi_H(Z_2) = 2\ln(\frac{1+\zeta_{Z_2}^{\ 2}}{2}) \tag{25}$$

$$\zeta_z = (1-\frac{16Z}{L})^{0.25} \tag{26}$$

While L=0, indicating a neutral state, $\Psi_m(Z_r) = \Psi_H(Z_1) = \Psi_H(Z_2) = 0$. Then, iterative calculation is carried out to correct H (Eqs. 27–29):

$$U_f^* = \frac{kU_r}{\ln(Z_r / z_{om}) - \Psi_m(Z_m)} \tag{27}$$

$$r_a^* = \frac{\ln(\frac{Z_2}{Z_1}) - \Psi_H(Z_1) - \Psi_H(Z_2)}{kU_f^*} \tag{28}$$

$$H = \frac{\rho_{air}C_p dT}{r_a^*} \tag{29}$$

Several iterations were carried out until the value of H was stable. Then, Eq. 1 was used to calculate $\lambda$ET. However, it should be noted that all the estimated energy component was an instantaneous value including latent heat; therefore, the concept of the evaporation fraction ($\Lambda$) was used to temporally scale-up from the instantaneous value to the daily ET. The evaporation fraction was defined as the ratio of latent heat to available energy (e.g., $R_n$-G) (Eq. 30). Several studies have indicated that the evaporation fraction can be regarded as constant throughout the day (Crago, 1996); therefore, the daily ET can be calculated as follows:

$$\Lambda = \frac{\lambda ET}{R_n - G} \tag{30}$$

$$ET_{daily} = \frac{\Lambda(R_{daily} - G_{daily})}{\lambda} \tag{31}$$

where $ET_{daily}$, $R_{daily}$, and $G_{daily}$ are the daily evapotranspiration, daily net radiation, and daily soil heat flux, respectively. Finally, the daily ET value was calculated. More details about SEBAL can be found in Bastiaanssen et al. (1998a).

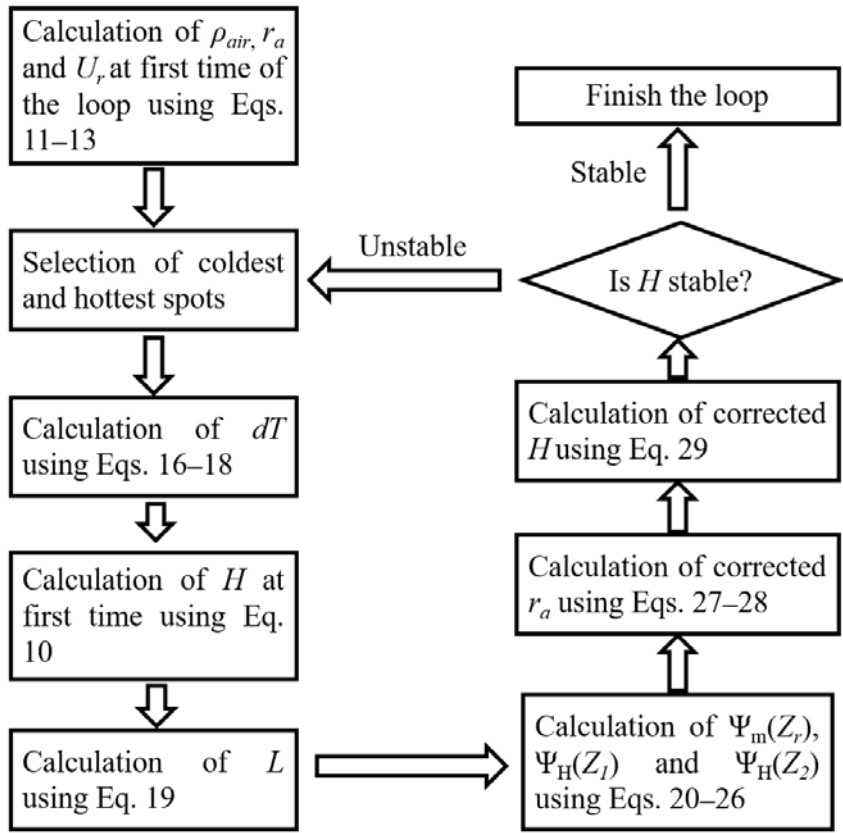

**Figure A1.** A flowchart of the calculation of sensible heat flux using Monin–Obkhov Similarity Theory (MOST).

**Appendix 2:Ratio of interpolated pixels of land surface temperature data**

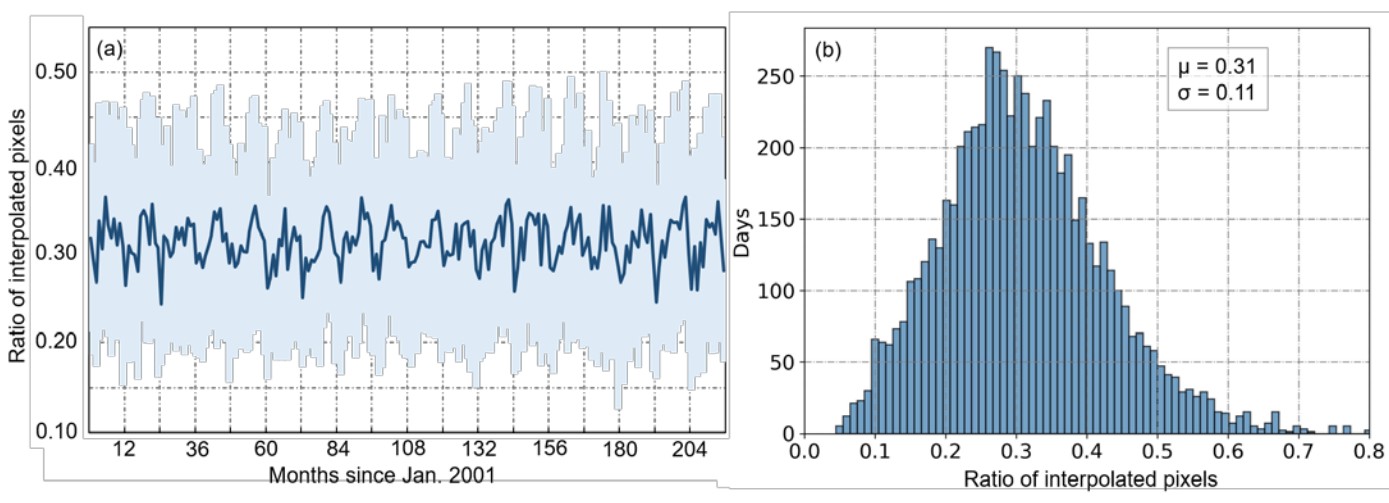

**Figure A2.** Ratio of interpolated pixels of land surface temperature (MOD11) data: (a) time series of interpolated pixels per month over 2001-2018 and (b) histogram of ratio of interpolated pixels.

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
