# Peer review of "Long time series of daily evapotranspiration in China based on the SEBAL model and multisource images and validation"

_Earth System Science Data, 2020_

## Referee Comment (RC1) · Anonymous Referee #1 · 29 Dec 2020

The authors have done a terrific job on this important contribution assessing actual ET over China land cover types using a long time series (2001–2018). They have provided a remarkable and complete and current assessment of the literature as well as provide the computational component in detail. They have compared an ET product based on SEBAL and multisource images for ET estimated using MOD16 data. They conducted a comprehensive validation of the product and compared its performance under different environmental conditions in China. They conclude that the ET product generated using SEBAL showed a good performance in China. They also provide next steps and explain the reasons for having these next steps. For example, nicely stated: "the improvement of the SEBAL algorithm will be the focus of follow-up research. Moreover,

the 1 km spatial resolution of the SEBAL ET product cannot meet the requirements of more detailed research. Due to the difficulty of simultaneously satisfying the requirements for the spatial and temporal resolutions of remote sensing data, the fusion of multiple sources of remote sensing data may be the most effective way to improve the spatiotemporal resolution of daily ET products." This provides the reader with a good understanding of what is still lacking/needed after this research which depicts four major contributions to the literature.

I have noticed only a few editorial corrections that need to be made, such as on pg. 25, the word "to" was left out "Compared the widely used MOD16 ET data,..." I have only one question that I would like to see them address in the text. Figure 11d shows SEBAL as a bi-modal curve whereas the summary histogram (Sebal in red) shape shows a low and wider distribution than MOD16 (blue). I'm wondering why SEBAL data is in fact bi-modal and the reasons for this need to be discussed. This is the only minor change I have other than to read it carefully for editorial mistakes.

Here is their own summary of their findings: Compared to flux tower observational data, the r-value of the SEBAL ET reached 0.79 for 9896 samples. Based on observational data from eight flux towers from 2003 to 2010, the ET datasets estimated using SEBAL and MOD16 were validated at the 8-day scale for different land cover types, climate zones, terrain types, and seasons. The results showed that SEBAL performed best in the conditions of forest cover (rRMSE = 38.08%), subtropical zones (rRMSE = 32.32%), hilly terrain (rRMSE = 32.32%), and the summer season (rRMSE = 36.56%), respectively, and performed worst in the conditions of grassland 520 cover (rRMSE = 52.63%), warm-temperate zones (rRMSE = 53.95%), plain terrain (rRMSE = 53.95%), and the winter season (rRMSE = 66.92%), respectively. Based on flux tower observational data and hydrological observational data, the ET estimated by SEBAL and MOD16 were validated at the point-scale and basin-scale. The results showed that, at the point-scale, the accuracy of SEBAL was 7.77 mm/8 d for the RMSE, 44.91% for the rRMSE, and 0.85 for the r-value. Overall, the SEBAL ET is higher than the MOD16 ET:

for 84.07% of the total area of China, the SEBAL ET showed higher values. Additionally, the SEBAL ET is closer to the in-situ measured ET in most conditions. Compared the widely used MOD16 ET data, the SEBAL ET product showed a higher accuracy and temporal resolution, but it still has a daily error of 42.04% (0.92 mm/d) at the point-scale and a yearly error of 19.15% (91.39 mm/year) at the basin-scale.

---

## Referee Comment (RC2) · Anonymous Referee #2 · 5 Mar 2021

General comments:

The study presents an approach of estimating long-term time series of daily ET in China by using the SEBAL model. In the current form, the manuscript lacks the literature to justify the need for the current study and several critical information related to SEBAL processing and ET validation. For example, there is almost no study reported in the introduction section that was conducted in China. There are several studies that used SEBAL and other surface energy balance (SEB) based models to estimate ET at a field and regional scales across different land covers and climates in China. Also, the authors did not report the critical information in the methods section such as the

selection of hot and cold pixels for the SEBAL processing. This is one of the main steps for the SEBAL model processing and the results may vary based on the different approaches applied (manual selection or automated selection). In addition, for the pixel-scale validation, the authors missed to report the quality of flux tower data and any approaches (e.g. constant Bowen-ratio, residual LE closure, . . .) applied to close the energy balance. These details are very basics and the core for any study related to SEB-based ET estimations. Without this critical information, it's difficult to warrant the validity of ET estimated from the current study.

Specific comments:

Section 2.2: lengthy model description. . .move it to appendix

Line 190-195: explain the gap-filling (spatial and temporal) process for pixels impacted with cloud

Line 196: any modification applied to MOD11 band for Ts adjustment?

Fig 3: could be moved to appendix

Line 223: describe the quality of flux tower data and any filtering applied to remove bad observations

Section 2.4.1: validation with flux tower and water balance would suffice

Fig 4:   add the time series plots as well. . ..provide more information for monthly/seasonal/annual variations

Fig 5: any obvious reason for ET underestimation for higher ET rates from SEBAL (for all land covers)?

Fig 8: discuss the seasonal overestimation/underestimation from SEBAL. . ..what are the primary driving factors?

Section 4.2.1: this section doesn't explain the quality of input data for the current study.

The QA/QC of input data is fundamental for ET modeling but this information is missing. The reference cited in line 425 is related to GPP…..not relevant to conclude that the quality of GMAO data was not accurate enough for ET modeling.

Section 4.2.2: this section is not discussing about the quality of flux tower data included in the current study…mostly literature…..not helpful to link with the results reported

Line 432: report the error from the flux towers considered in this study

Line 443: report the footprint of flux towers used in this study…this is critical for point-scale validation

Line 445: any explanation about overestimation during winter? Also, discuss the SE-BAL overestimation at lower ET rates and underestimation at higher ET rates in Figure 7 and Figure 8

Section 4.3.2: not relevant to discuss the results from SEBAL, could be removed

Line 490-495: report the spatial (tiles/basins) and temporal (study years) variation of hot and cold pixels …..would be helpful to link with the reported results

Line 490-499: the sources of errors related to H estimation can be evaluated with instantaneous H from flux tower…this would help to identify where the errors are coming from (maybe from modeled Rn and G too)…..along with the quality of input data and flux tower data

———————————————

---

## Author Comment (AC1) · 17 Mar 2021

Thank you for the positive appreciation of our work. We will make the following changes to the suggestions:

1. I have noticed only a few editorial corrections that need to be made, such as on pg. 25, the word "to" was left out "Compared the widely used MOD16 ET data..."

Reply: "Compared the widely used MOD16 ET data ..."(Line 543, Page 25) will change to "Compared to the widely used MOD16 ET data ...". Moreover, we will further to check the editorial mistakes in the whole manuscript as well.

2. I would like to see them address in the text. Figure 11d shows SEBAL as a bi-modal curve whereas the summary histogram (Sebal in red) shape shows a low and wider distribution than MOD16 (blue). I'm wondering why SEBAL data is in fact bi-modal and the reasons for this need to be discussed. This is the only minor change I have other than to read it carefully for editorial mistakes.

Reply: According to your suggestion, the following paragraph will add to discuss the question: "Regarding the distribution of ETSEBAL, a bi-modal curve with the boundary of ~500mm was shown in the Fig. 11d, it was likely contributed by the misestimation of part of regions. The ETSEBAL map was divided to two parts with 500 mm as threshold (Fig. 11f, g), the part of ETSEBAL below 500 mm was distributed in the Northwest China whereas the part of ETSEBAL over 500 mm was distributed in the southeast. It should be noted that the vegetation cover in northwest of China are mainly grassland and a small part of cropland (Fig. 11h), and SEBAL was underestimated the ET of grassland and cropland (Section 3.1). In contrast, ETSEBAL showed slightly overestimation of forest which is the main land cover types in southeast of China. Therefore, the part which ET should have been distributed around ~500 mm was underestimated or overestimated, and thus caused the bi-modal curve."

Please also note the supplement to this comment:
https://essd.copernicus.org/preprints/essd-2020-345/essd-2020-345-AC1-supplement.pdf

—————————————————————

[Figure]

**Fig. 1.**

**Supplement:**

**Reply for Anonymous Referee #1:**

Thank you for the positive appreciation of our work. We will make the following changes to the suggestions:

**1. I have noticed only a few editorial corrections that need to be made, such as on pg. 25, the word "to" was left out "Compared the widely used MOD16 ET data..."**

**Reply:** "Compared the widely used MOD16 ET data ..."(Line 543, Page 25) will change to "Compared to the widely used MOD16 ET data ...". Moreover, we will further to check the editorial mistakes in the whole manuscript as well.

**2. I would like to see them address in the text. Figure 11d shows SEBAL as a bi-modal curve whereas the summary histogram (Sebal in red) shape shows a low and wider distribution than MOD16 (blue). I'm wondering why SEBAL data is in fact bi-modal and the reasons for this need to be discussed. This is the only minor change I have other than to read it carefully for editorial mistakes.**

**Reply:** According to your suggestion, the following paragraph will add to discuss the question (Section 3.4, Page 18, Lines 340 - 346):

"Regarding the distribution of $ET_{SEBAL}$, a bi-modal curve with the boundary of ~500mm was shown in the Fig. 11d, it was likely contributed by the misestimation of part of regions. The $ET_{SEBAL}$ map was divided to two parts with 500 mm as threshold (Fig. 11f, g), the part of $ET_{SEBAL}$ below 500 mm was distributed in the Northwest China whereas the part of $ET_{SEBAL}$ over 500 mm was distributed in the southeast. It should be noted that the vegetation cover in northwest of China are mainly grassland and a small part of cropland (Fig. 11h), and SEBAL was underestimated the ET of grassland and cropland (Section 3.1). In contrast, $ET_{SEBAL}$ showed slightly overestimation of forest which is the main land cover types in southeast of China. Therefore, the part which ET should have been distributed around ~500 mm was underestimated or overestimated, and thus caused the bi-modal curve."

[Figure]

**Figure 11.** A comparison between the SEBAL and MOD16 models. (a) Distribution of annual average $ET_{SEBAL}$; (b) Distribution of annual average $ET_{MOD}$; (c) Distribution of the difference between SEBAL and MOD16 (ETSEBAL – ETMOD); (d) Histogram of annual average $ET_{SEBAL}$ and $ET_{MOD}$; (e) Histogram of the relative difference between SEBAL and MOD16 (($ET_{SEBAL}$ – $ET_{MOD}$)/$ET_{MOD}$); (f) Map of $ET_{SEBAL}$ over 500 mm; (g) Map of $ET_{SEBAL}$ below 500 mm; (h) Land cover in China.

---

## Author Comment (AC2) · 17 Mar 2021

Reply for Anonymous Referee #2:

Thank you for the positive appreciation of our work. We will make the following changes according to your good suggestions:

General comments:

1. The study presents an approach of estimating long-term time series of daily ET in China by using the SEBAL model. In the current form, the manuscript lacks the

literature to justify the need for the current study and several critical information related to SEBAL processing and ET validation. For example, there is almost no study reported in the introduction section that was conducted in China. There are several studies that used SEBAL and other surface energy balance (SEB) based models to estimate ET at a field and regional scales across different land covers and climates in China.

Response: We have supplemented several literatures about performance of SEB-based model in China, and analyzed the need for this study (Section 1, Introduction, Page 3, Lines 75 - 91 in new version). Thank you for your help and good suggestions in improving our manuscript.

2. Also, the authors did not report the critical information in the methods section such as the selection of hot and cold pixels for the SEBAL processing. This is one of the main steps for the SEBAL model processing and the results may vary based on the different approaches applied (manual selection or automated selection).

Response: In the process of SEBAL ET generating, the hot and cold pixels were selected automatedly by following the certain rules which referred to previous studies, and these parts were supplemented (Appendix, Pages 27 - 28, Lines 580 - 588 in new version). Thank you for your help and good suggestions in improving our manuscript.

3. In addition, for the pixel-scale validation, the authors missed to report the quality of flux tower data and any approaches (e.g. constant Bowen-ratio, residual LE closure...) applied to close the energy balance. These details are very basics and the core for any study related to SEB-based ET estimations. Without this critical information, it's difficult to warrant the validity of ET estimated from the current study.

Response: Regarding the quality of flux tower data, we have filtered and corrected the flux tower measured data, first, we selected the high-quality data with Energy Balance Closure Ratio (ECR) is more than 80%, and further used Bowen Ratio energy balance method to correct the selected data, these parts were supplemented (Section 2.3.1, Page 6, Lines 144 - 154 in new version). Thank you for your help and good suggestions

in improving our manuscript.

Specific comments:

1. Section 2.2: lengthy model description...move it to appendix

Response: These parts were moved to appendix (Pages 25 - 30, Lines 530 - 625 in new version). Thank you for your help and good suggestions in improving our manuscript.

2. Line 190-195: explain the gap-filling (spatial and temporal) process for pixels impacted with cloud

Response: Regarding the MODIS data used for SEBAL input (MOD11. MOD13 and MCD43), we have filled the missed or unreliable (caused by cloud or other reasons) pixels and the methods were referred to previous studies (Section 2.2, Page 3, Lines 125 - 130 in new version). Regarding the MOD16 ET data, the missed or unreliable pixels were not used for the comparison with SEBAL ET and not filled. Thank you for your help and good suggestions in improving our manuscript.

3. Line 196: any modification applied to MOD11 band for Ts adjustment?

Response: In this study, the daytime surface temperature in MOD11 was used for SEBAL input, the data have not been modified except dimensional conversion (scale factor is 0.02) and gap-filling (Section 2.2, Page 5, Lines 125 - 130 in new version). Thank you for your help and good suggestions in improving our manuscript.

4. Fig 3: could be moved to appendix.

Response: This figure was moved to appendix. Thank you for your help and good suggestions in improving our manuscript. (Appendix, Page 30in new version)

5. Line 223: describe the quality of flux tower data and any filtering applied to remove bad observations

Response: In this study, we selected the high-quality data with Energy Balance Closure Ratio (ECR) is more than 80%, and further used Bowen Ratio energy balance method to correct the selected data, these parts were supplemented (Section 2.3.1, Page 6, Lines 144 - 154 in new version). Thank you for your help and suggestions in improving our manuscript.

6. Section 2.4.1: validation with flux tower and water balance would suffice.

Response: The ET obtained from flux tower and water balance method could efficiently validate SEBAL ET. Moreover, the MOD16 ET product is one of widely used evapotranspiration dataset for water resources management and global change study, which also performs accurate to some extent. In this study, the comparison of SEBAL ET and MOD16 ET was conducted to judge if the further improvement was found in SEBAL ET (Section 2.4.1, Page 8, Lines 187 - 189 in new version). Thank you for your help and good suggestions in improving our manuscript.

7. Fig 4: add the time series plots as well...provide more information for monthly/seasonal/annual variations

Response: The time series plots of ET in the flux tower stations were added as Fig .5, and we further described the ET variation characteristics in time series (Section 3.1, Pages 9 - 10, Lines 226 - 232 in new version). Thank you for your help and good suggestions in improving our manuscript.

8. Fig 5: any obvious reason for ET underestimation for higher ET rates from SEBAL (for all land covers)?

Response: The reason of underestimation of SEBAL ET at higher ET was discussed in addition (Section 4.2.1, Page 21, Lines 386 - 400 in new version), which may cause by the saturation issue of optical sensor. For example, in the dense vegetation covers, the vegetation index (e.g., NDVI) was likely underestimated and can not accurately characterize vegetation status, therefore, soil heat flux will be overestimated according

to Eq. 9 (in appendix), and may further caused the sensible heat flux underestimation. Thank you for your help and good suggestions in improving our manuscript.

9. Fig 8: discuss the seasonal overestimation/underestimation from SEBAL...what are the primary driving factors?

Response: We discussed the reason of the seasonal overestimation/underestimation from SEBAL in addition (Section 4.2.1, Page 21, Lines 386 - 400 in new version). For example, the obvious overestimation in spring and summer may cause by gap-filling of unreliable pixels, spring and summer have the relatively frequent precipitation, which causes more unreliable pixels due to the cloud, and these pixels value were finally replaced by gap-filling of nearest date pixel value, therefore, the modeled ET value of these pixels was close to that of nearest date without precipitation. Actually, due to the high air humidity in rainy day, the evaporation and transpiration are relatively less than that of nearest date (Ferreira and Cunha, 2020; Li et al., 2016). Moreover, it should be noted, due to the decrease of surface temperature after precipitation, the ET (both actual and modeled value) is also in a relatively low level (Cheng et al., 2020). This may explain the reason of obvious overestimation at lower ET rates in spring, summer and other pixels affected by cloud. For underestimation of SEBAL ET at higher ET rates, which may cause by saturation issue of optical sensor. Thank you for your help and good suggestions in improving our manuscript.

10. Section 4.2.1: this section doesn't explain the quality of input data for the current study. The QA/QC of input data is fundamental for ET modeling but this information is missing.

Response: In this study, the MODIS quality control (QC) file were used to distinguish the unreliable pixels of MODIS data (MOD11, MOD13 and MOD43) and then the gap-filling method were applied for fill or replace these unreliable pixels (Section 2.2, Page 5, Lines 125 - 130 in new version). Thank you for your help and good suggestions in improving our manuscript.

11. The reference cited in line 425 is related to GPP....not relevant to conclude that the quality of GMAO data was not accurate enough for ET modeling.

Response: These parts have been removed (Section 4.2.1, Page 20, Lines 425 - 426 in old version). Thank you for your help and good suggestions in improving our manuscript.

12. Section 4.2.2: this section is not discussing about the quality of flux tower data included in the current study...mostly literature.....not helpful to link with the results reported

Response: We have rewritten this section and further discussed the errors may cause by flux tower in this study (Section 4.2.2, Pages 21 - 22, Lines 405 - 426 in new version). Thank you for your help and good suggestions in improving our manuscript.

13. Line 432: report the error fr om the flux towers considered in this study

Response: We have rewritten Section 4.2.2. In this study, the Bowen ratio method (Eq. 3), which assuming that the residual of the energy balance is attributed to sensible and latent heat flux and assigning the missing energy flux to them, was used to enforce energy closure. Actually, this assumption is not very correct, which generally led the sensible and latent heat flux overestimation, which may could explain that the SEBAL ET was generally underestimated when compared to flux tower observed ET (Fig. 9). The same issue was found in regional-scale validation, due to the ignoring of $\Delta S$ in the water balance computation process (although it's really small), which could lead the regional ET overestimation and further caused SEBAL ET underestimation in validation (Fig. 10) (Section 4.2.2, Page 21, Lines 405 - 414 in new version). Moreover, the 1 km $\times$ 1 km area of pixel was used for matching the footprint of flux tower which was referred to the study of Velpuri et al. (2013), however, the footprint is not stable but varied with environment changed, e.g., vegetation height. Chen et al. (2012) reported that forest footprint has clear difference with grassland, the footprint of forest is much larger which is reached kilometer-scale. In fact, forest footprint may more matching with

the spatial resolution in this study. Therefore, it may explain that the SEBAL ET has the greatest performance in forest but worst performance in grassland. Compared to the study of Velpuri et al. (2013), the grassland also showed the worst remote sensing ET estimation in US when using flux tower data for validation at a kilometer-scale (Section 4.2.2, Page 22, Lines 415 - 425 in new version). Thank you for your help and good suggestions in improving our manuscript.

14. Line 443: report the footprint of flux towers used in this study...this is critical for pointscale validation

Response: In this study, the 1 km $\times$ 1 km of pixel was matched with flux footprint (Section 2.3.1, Page 6, Lines 152 - 154 in new version). And we further discussed the errors may cause by the footprint issue (Section 4.2.2, Page 22, Lines 415 - 425 in new version). Thank you for your help and good suggestions in improving our manuscript.

15. Line 445: any explanation about overestimation during winter? Also, discuss the SEBAL overestimation at lower ET rates and underestimation at higher ET rates in Figure 7 and Figure 8

Response: The contents in Section 3.2.4 showed that SEBAL ET has the highest error in winter (rRMSE = 66.92%), but the error did not show obvious underestimation or overestimation (MBE= -0.62 mm/8d). We further discussed the reason of the highest error in winter, which may cause by the low temperature and snow cover in winter (Section 4.2.1, Page 21, Lines 393 - 395 in new version). Moreover, we also discussed the possible reason that the SEBAL overestimation at lower ET rates and underestimation at higher ET rates (Section 4.2.1, Page 21, Lines 395 - 399 in new version). Thank you for your help and good suggestions in improving our manuscript.

16. Section 4.3.2: not relevant to discuss the results from SEBAL, could be removed.

Response: These parts have been removed (Section 4.3.2, Page 21, Lines 460 - 479 in old version). Thank you for your help and good suggestions in improving our

manuscript.

17. Line 490-495: report the spatial (tiles/basins) and temporal (study years) variation of hot and cold pixels....would be helpful to link with the reported results

Response: The hot and cold pixels selection error is one of the main causes of SEBAL model uncertainties. We further discussed the influence of temporal variation of hot and cold pixels to SEBAL. A study reported that the cold pixel performed more stable than hot pixel in time series, especially in winter, the hot pixel was highly varied may due to the similarity of NDVI over space, it could further explain the poor performance of SEBAL ET in winter (Section 4.3.2, Page 23, Lines 451 - 458 in new version). Thank you for your help and good suggestions in improving our manuscript.

18. Line 490-499: the sources of errors related to H estimation can be evaluated with instantaneous H from flux tower. this would help to identify where the errors are coming from (maybe from modeled Rn and G too)....along with the quality of input data and flux tower data.

Response: The source of ET estimation errors is a subject worthy of further study. In this paper, we referred to previous studies to further discussed this question: 'Besides sensible heat flux, the errors of SEBAL ET may derived from net radiation or soil heat flux as well (Li et al., 2017; Teixeira et al., 2009). For net radiation, which is computed using surface albedo and Stephen Boltzmann law (Eq. 2 in appendix), generally performed a relatively agreement with flux tower observed value, while soil heat flux, which computed using empirical formula related to net radiation and NDVI (Eq. 9 in appendix), has a poor performance (Li et al., 2017; Song et al., 2016). In the study of Li et al. (2017), soil heat flux estimation showed a clear overestimation in higher ET area, e.g., wetland, which may further cause the sensible and latent heat flux underestimation in higher ET rates. In the most SEB-based algorithms, the similar net radiation and soil heat flux estimation methods are used, and various sensible heat flux estimation methods are the main sources of the difference among

the various SEB-based algorithms. However, the causes of the net radiation and soil heat flux estimation errors have not been clearly discussed, e.g., the effect of satellite transmitted time or land cover types. These issues could be the focus of our follow-up research, for example, high frequency geostationary satellite and flux tower observations may be helpful for this research.' (Section 4.3.2, Page 23, Lines 462 - 471 in new version). However, due to the limited of flux tower data, we could not study the instantaneous energy component in this paper, and this object will be conducted in our follow-up research. Thank you for your help and good suggestions in improving our manuscript.

Please also note the supplement to this comment:
https://essd.copernicus.org/preprints/essd-2020-345/essd-2020-345-AC2-supplement.pdf

---

## Referee Report (RR1)

General comments:

The manuscript introduces a newly generated ET data set with 1km spatial resolution and daily temporal resolution over China based on the SEBAL model. Given that this manuscript is a contribution to ESSD, with a focus on the newly provided data product, I am missing information on and/or discussion of critical issues such as the selection of the extreme pixel values for the SEBAL model, the amount of missing data in the LST time series, the derivation of ET from the water balance. The comparison with MOD16 is a bit lengthy, and much of the information could be put together in tables, rather than listing all performance scores for each vegetation class, terrain class etc in the running text.

Since I am not a native speaker myself, I do not comment on language at all, but the manuscript needs rigorous english proofreading.

Major comments:
1) The selection of the pixels that define the extreme hot and cold conditions is a critical step in the application of the SEBAL model. In the Appendix the authors describe their routine for the extreme pixel selection: they select a single hot and cold pixel over the MODIS scene. I am missing a discussion on the justification of this approach; given that a single MODIS scene covers an area of 1200x1200 km, with differences in elevation, weather conditions etc., I don't think that two extreme pixels coming maybe from points very far apart from each other could be related to each other in a reasonable way. Because in the SEBAL method it is assumed that changes in LST are mainly due to the evaporative cooling effect, rather than elevation variation, shadows etc. I would be interested to see an analysis showing the sensitivity of the extreme LST pixels to different selection methods.

2) If I understand the authors correctly, they calculate a yearly water balance ET for nine primary water resources divisions. My first question here would be whether they selected hydrological years or calendar years? Second, from my own experience with ET derived from the water balance, an averaging period of a year is not enough to ensure the assumption of ignorable storage changes. However, I have no experience with such large basins. Figure 10 shows that the variation of ETwb is quite significant for some of the basins. I would therefore encourage the authors to discuss their approach and its implications.

3) LST data availability is often a major limitation when applying LST-based ET algorithms. The statement "it should be noted that there are several missing or unreliable pixels in MODIS images" is a bit vague, in my opinion. I would prefer some quantification of the share of valid to invalid pixel values, e.g. in the form of a percentage of valid data points in the time series per pixel, or a table with similar information further categorized into seasons, etc. The authors apply a very simple data imputation method and it would be interesting for the reading to know how much of the modelled ET values are based on these interpolated LST data. Even a flag in the data set could be considered.

4) For the comparison with the EC data, it would be interesting to also include the other energy balance components, sensible and ground heat flux and net radiation.

5) Given that only eight EC towers are available for model evaluation, I am wondering whether a differentiation of model performance according to land cover (three types) but even more climate zones (five zones) and terrain classes (four classes) makes sense. I am not sure how well suited the available data are to draw general conclusions on the performance in the different climate zones, etc.

Additional comments:

P. 2, line 38: what are traditional methods in this context? Remote sensing models rely on very traditional approaches (Penman-Monteith or surface energy balance residual models are very traditional approaches.

P. 3, line 63: In my opinion the classification of ET models into SEB and SEF models is a bit subjective. The Penman-Monteith equation e.g. is also physically-based (as SEB approaches).

P. 3, line 69: The authors state that the temporal resolution of eight days is not sufficient for search on water resources management? How do the authors come to this conclusion and what temporal resolution would be sufficient?

P. 5, line 125: see major comment 3)

P. 6, line 141: The authors state that the EC method measures ET using the covariance between wavpor and heat fluxes. This is wrong! The EC method measures the covariance of the vertical wind velocity (!) and concentration of the entity of interest.

P. 7, line 163: see major comment 2)

P. 8, line 180: RMSE is not suited to describe model bias.

P. 9, line 225: The authors conclude that "ET_SEBAL is relatively reliable for daily-scale application". I am wondering how they justify that statement and if they have some references to define what a relatively reliable model performance is.

P. 11, line 240: see major comment 5)

P.21, line 390: The authors state that a decrease in surface temperature corresponds to a reduced evapotranspiration. I think this needs rephrasing because in general low surface temperature at similar meteorological forcing would indicate that more of the available energy is dissipated via ET than sensible heat flux.

P. 22, line 428: I am wondering why the authors decided for the described upscaling method, if they explain in this section why other methods would be preferable.

P. 23, line 444: see major comment 4)

P. 23, line 454: What do the authors mean by " a low domain size" in this sentence?

P. 26, line 538: the use of the arrows for indicating up- and downwelling radiation is inconsistent between equations and the text.

P. 27: line equation (12) is true for neutral conditions only.

P. 28, line 580: see major comment 1)

P. 28: some of the equations (20) to (28) are redundant.

Zenodo homepage: The authors state "The products were evaluated using the eight flux towers observation data for point validation and water balance method for regional validation and showed R value of 0.79 and 0.88, respectively, which indicated the products have a great performance". In my opinion a rRMSE of > 40 % might maybe not indicate great performance.

Zenodo homepage: the coordinate system should be stated in the text

---

## Referee Report (RR2)

The authors addressed most of my concerns. Below I list a few further comments:

**Comment 1:**

The author reply to my comment concerning the EC measurement principle: "We have rewritten this sentence by referring the paper of Wang et al. (2012): "The eddy covariance method measures  $\lambda ET$  from the covariance of the heat and moisture fluxes, respectively, with vertical velocity using rapid response sensors at frequencies typically equal to or greater than 10 Hz" (Page 7, Lines 147-148)."

This is again wrong. And the authors use the paper Wang and Dickinson 2012 as a reference but cite it incorrectly. The original sentence from the paper reads: *"The EC technique measures H and IE from the covariance of the heat and moisture fluxes, respectively, with vertical velocity using rapid response sensors at frequencies typically equal to or greater than 10 Hz."*

The covariance between heat fluxes and vertical wind velocity defines the sensible heat flux, while the covariance between the moisture flux and the vertical wind velocity defines the latent heat flux. That the authors decided to cite this sentence only partially (leaving out the sensible heat flux) is very critical in my view. This repeated wrong definition of the EC method, makes me doubt the author's understanding of the EC theory.

**Comment 2:**

The authors replied very detailedly to my comment on the differentiation into different climate zones, terrain types etc. by referencing other papers that used a similar number of EC stations for assessing model performance. In my opinion, this is still not convincing since e.g. the class "cropland" includes a single site while croplands exhibit very different ET rates and dynamics based on the cultivated crop. The same will hold true e.g. for climate types. In my opinion the authors should at least state that the low number of stations per class (land cover, terrain type etc.) might reduce the validity of these findings.

Comment 3:

The equations (26) to (28) are still redundant.

Comment 4:

The manuscript still needs english proofreading.

---

## Author Response (AR2)

**Ref.: Dr. No. ESSD-2020-345**

**Title**: Long time series of daily evapotranspiration in China based on the SEBAL model and multisource images and validation

**Author**: Minghan Cheng, Xiyun Jiao, Binbin Li, Xun Yu, Mingchao Shao, Xiuliang Jin

**Research Paper**

**Earth System Science Data**

**Cover letter**

**Dear Editor and Reviewers**

I am submitting here a manuscript entitled "*Long time series of daily evapotranspiration in China based on the SEBAL model and multisource images and validation*". We submitted this manuscript in November 2020. Three reviewers gave us good advices. First we would like to thank the reviewers for their constructive and helpful suggestions and improvements to our manuscript (**ESSD-2020-345**). We revised the manuscript by following the suggestions of the reviewers. Our response to each suggestion or comment are given one by one in the following Pages of this letter. For details, please refer to the responses as follows (Reviewer comments are in black font, responses are in blue or red font)

Looking forward to your favorable decision.

Thanks too much.

With best regards,

Minghan Cheng and co-authors

**Responses to Reviewers**

**Reviewer 2#:**

The authors have addressed most of my previous comments (mostly in the methods section). A few further modification/classifications suggested in the revised version:

1. Line 399: .....caused the latent heat flux underestimation?

**Response**: Thank you for your help and suggestions in improving our manuscript. It has been revised (**Line 404, Page 21**).

2. Line 410-411: if the assumption of constant Bowen ratio was not correct, the authors must have used the correct/suitable method for closing the energy balance. Please provide more clarification or rephrase the sentence.

**Response**: Thank you for your help and suggestions in improving our manuscript. In this study, the eddy covariance system measured value was filtered and corrected. First, the data with Energy Balance Closure Ratio (ECR, Eq. 2) less than 80% were not selected for validation, and then, the remaining data with ECR more than 80% were corrected by using Bowen Ratio energy balance correction (Eq. 3) (**Lines 152-156, Page 7**).

**Reviewer 3#:**

**General comments:**

The manuscript introduces a newly generated ET data set with 1km spatial resolution and daily temporal resolution over China based on the SEBAL model. Given that this manuscript is a contribution to ESSD, with a focus on the newly provided data product, I am missing information on and/or discussion of critical issues such as the selection of the extreme pixel values for the SEBAL model, the amount of missing data in the LST time series, the derivation of ET from the water balance. The comparison with MOD16 is a bit lengthy, and much of the information could be put together in tables, rather than listing all performance scores for each vegetation class, terrain class etc in the running text. Since I am not a native speaker myself, I do not comment on language at all, but the manuscript needs rigorous English proofreading.

**Response**: First we would like to thank the reviewers for their constructive and helpful

suggestions and improvements to our manuscript. Our response to each suggestion or comment are given one by one in the following Pages of this letter. MOD16 products is one of widely used ET dataset, so we selected it as a typical for comparing with the generated SEBAL ET product in this paper. If the SEBAL ET showed a comparable performance with MOD16 or even better, which could indicate the SEBAL ET have an acceptable accuracy and can be used for related studies.

**Major comments:**

1. The selection of the pixels that define the extreme hot and cold conditions is a critical step in the application of the SEBAL model. In the Appendix the authors describe their routine for the extreme pixel selection: they select a single hot and cold pixel over the MODIS scene. I am missing a discussion on the justification of this approach; given that a single MODIS scene covers an area of 1200 x 1200 km, with differences in elevation, weather conditions etc., I don't think that two extreme pixels coming maybe from points very far apart from each other could be related to each other in a reasonable way. Because in the SEBAL method it is assumed that changes in LST are mainly due to the evaporative cooling effect, rather than elevation variation, shadows etc. I would be interested to see an analysis showing the sensitivity of the extreme LST pixels to different selection methods.

**Response**: Thank you for your help and suggestions in improving our manuscript. The hot/cold pixel selection method in this study was referred to Long et al. (2011), which has been referred in many studies. Domain size (defined as the actual size of the modeling domain/satellite imagery being used) is an important effect for hot/cold pixel selection, in the study of Long et al. (2011), this issue has been deeply discussed. In this study, we further compared the different domain sizes' performance in ET estimation (**Fig. 12 Lines 475, Page 24**), overall, the domain size employed in this study (1200 km × 1200 km) performed an acceptable accuracy (**Lines 457-475, Pages 23-24**).

2. If I understand the authors correctly, they calculate a yearly water balance ET for nine primary water resources divisions. My first question here would be whether they selected hydrological years or calendar years? Second, from my own experience with ET derived from the water balance, an averaging period of a year is not enough to ensure the assumption of ignorable storage changes. However, I have no experience with such large basins. Figure 10 shows that the variation of ETwb

is quite significant for some of the basins. I would therefore encourage the authors to discuss their approach and its implications.

**Response**: Thank you for your help and suggestions in improving our manuscript. We used average $ET_{WB}$ over multiple years instead of in one year for regional-scale validation (**Lines 320-327, Page 17**). $\Delta S$ over multiple years can be ignored (Liu et al., 2016; Senay et al., 2011) (**Lines 174-176, Page 8**). The $ET_{WB}$ was calculated using observed data recorded in calendar years.

3. LST data availability is often a major limitation when applying LST-based ET algorithms. The statement 'it should be noted that there are several missing or unreliable pixels in MODIS images' is a bit vague, in my opinion. I would prefer some quantification of the share of valid to invalid pixel values, e.g. in the form of a percentage of valid data points in the time series per pixel, or a table with similar information further categorized into seasons, etc. The authors apply a very simple data imputation method and it would be interesting for the reading to know how much of the modelled ET values are based on these interpolated LST data. Even a flag in the data set could be considered.

**Response**: Thank you for your help and suggestions in improving our manuscript. We supplemented two figures to describe the ratio of interpolated pixels of land surface temperature (MOD11) data (**Lines 640-643, Page 31**). Fig. A2a describes the time series of interpolated pixels per month over 2001-2018 and Fig.A2b describes the histogram of ratio of interpolated pixels.

4. For the comparison with the EC data, it would be interesting to also include the other energy balance components, sensible and ground heat flux and net radiation.

**Response**: Thank you for your help and suggestions in improving our manuscript. The source of ET estimation errors is a subject worthy of further study. In this paper, we referred to previous studies to further discussed this question: *'Besides sensible heat flux, the errors of SEBAL ET may derived from net radiation or soil heat flux as well (Li et al., 2017; Teixeira et al., 2009). For net radiation, which is computed using surface albedo and Stephen Boltzmann law (Eq. 2 in appendix), generally performed a relatively agreement with flux tower observed value, while soil heat flux, which computed using empirical formula related to net radiation and NDVI (Eq. 9 in appendix), has a poor performance (Li et al., 2017; Song et al., 2016). In the study of Li et al.*

*(2017), soil heat flux estimation showed a clear overestimation in higher ET area, e.g., wetland, which may further cause the sensible and latent heat flux underestimation in higher ET rates. In the most SEB-based algorithms, the similar net radiation and soil heat flux estimation methods are used, and various sensible heat flux estimation methods are the main sources of the difference among the various SEB-based algorithms. However, the causes of the net radiation and soil heat flux estimation errors have not been clearly discussed, e.g., the effect of satellite transmitted time or land cover types. These issues could be the focus of our follow-up research, for example, high frequency geostationary satellite and flux tower observations may be helpful for this research.'* (**Section 4.3.2, Page 24, Lines 476 - 486**). However, due to the limited of flux tower data, we could not study the instantaneous energy component in this paper, and this object will be conducted in our follow-up research. Thank you for your help and good suggestions in improving our manuscript.

5. Given that only eight EC towers are available for model evaluation, I am wondering whether a differentiation of model performance according to land cover (three types) but even more climate zones (five zones) and terrain classes (four classes) makes sense. I am not sure how well suited the available data are to draw general conclusions on the performance in the different climate zones, etc.

**Response**: Thank you for your help and suggestions in improving our manuscript. EC flux tower has been proved that could be used for regional ET validation, even better than most of other approaches (Wang et al., 2012). For example, Hu et al. (2015) used 15 flux towers to validate the performance of MOD16 and LSA-SAF MSG evapotranspiration products over Europe; Aguilar et al. (2018) used five flux towers to validate MOD16 ET product over Northwestern Mexico; Ramoelo et al. (2014) used two flux towers to validate MOD16 ET products over parts area of South Africa; Yang et al. (2017) used eight towers to validate GLEAM ET products over China; Li et al. used 12 towers to validate GLEAM and GLDAS ET products over China; Kim et al. (2012) used 20 towers to validate MOD16 products over Asia. In general, the density of flux towers in this study (eight towers in China) is comparable with previous studies, even better than some of them.

Moreover, the performance assessment of the product under different conditions (climate zones,

ecosystems and terrain) could make the results more comprehensive (Velpuri et al., 2013), in this study, the eight towers basically cover most of climate zones, ecosystems and terrain in China, and the observed period is long-time-series (each tower both have more than 1000 samples with total available samples of 9896). Therefore, the validation of MOD16 in this study is convincingness to some extent.

**References:**

Wang, K. and Dickinson, R.E., 2012. A review of global terrestrial evapotranspiration: Observation, modeling, climatology, and climatic variability. Reviews of Geophysics, 50(2).

Hu, G., Jia, L. and Menenti, M., 2015. Comparison of MOD16 and LSA-SAF MSG evapotranspiration products over Europe for 2011. Remote Sensing of Environment, 156: 510-526

Aguilar, A. et al., 2018. Performance Assessment of MOD16 in Evapotranspiration Evaluation in Northwestern Mexico. Water, 10(7).

Ramoelo, A. et al., 2014. Validation of Global Evapotranspiration Product (MOD16) using Flux Tower Data in the African Savanna, South Africa. Remote Sensing, 6(8).

Yang, X., Yong, B., Ren, L., Zhang, Y. and Long, D., 2017. Multi-scale validation of GLEAM evapotranspiration products over China via ChinaFLUX ET measurements. International Journal of Remote Sensing.

Kim, H.W., Hwang, K., Mu, Q., Lee, S.O. and Choi, M., 2012. Validation of MODIS 16 global terrestrial evapotranspiration products in various climates and land cover types in Asia. KSCE Journal of Civil Engineering, 16(2).

Velpuri, N.M., Senay, G.B., Singh, R.K., Bohms, S., & Verdin, J.P., 2013. A comprehensive evaluation of two MODIS evapotranspiration products over the conterminous United States: Using point and gridded FLUXNET and water balance ET. Remote Sensing of Environment.

**Additional comments:**

1. P. 2, line 38: what are traditional methods in this context? Remote sensing models rely on very traditional approaches (Penman-Monteith or surface energy balance residual models are very

traditional approaches.)

**Response**: Thank you for your help and suggestions in improving our manuscript. The traditional methods indicate the methods based on point-scale or small-area-scale analysis, such as lysimeter and eddy covariance. In order to make it, this sentence was rephased: "*However, the methods for the estimation of ET based on point-scale or small-area-scale analysis, such as lysimeter and eddy covariance, cannot meet the requirement of global climate change research and regional water resource management*" (**Page 2, Lines 38 - 40**).

2. P. 3, line 63: In my opinion the classification of ET models into SEB and SEF models is a bit subjective. The Penman-Monteith equation e.g. is also physically-based (as SEB approaches).

**Response**: Thank you for your help and suggestions in improving our manuscript. In the revised version of manuscript, we divided into three types according to their mechanism: those based on surface energy balance residual (SEBR), those based on semi-empirical formulas (SEFs) and statistic methods by referring the paper of Wang et al. (2012) and Zhang et al. (2016). The P-M and P-T equation, which is partly physical, were divided into semi-empirical formula method. Moreover, we add a category - statistic methods to avoid confusion with SEF-based method (**Pages 2-3, Lines 47 - 66**).

3. P. 3, line 69: The authors state that the temporal resolution of eight days is not sufficient for search on water resources management? How do the authors come to this conclusion and what temporal resolution would be sufficient?

**Response**: Thank you for your help and suggestions in improving our manuscript. It indicates that higher temporal resolution could use for finer water resources management, e.g., irrigation regime making. In order to avoid misunderstanding, we deleted this sentence (**Page 3, Line 71**).

4. P. 5, line 125: see major comment 3

**Response**: Thank you for your help and suggestions in improving our manuscript. We supplemented an appendix to show the ratio of interpolated pixels of land surface temperature (MOD11) data (**Pages 6, Lines 133-135 in new version**). The details were described in the response of major comment 3.

5. P. 6, line 141: The authors state that the EC method measures ET using the covariance between vapor and heat fluxes. This is wrong! The EC method measures the covariance of the vertical wind velocity (!) and concentration of the entity of interest.

**Response**: Thank you for your help and suggestions in improving our manuscript. We have rewritten this sentence by referring the paper of Wang et al. (2012): "*The eddy covariance method measures λET from the covariance of the heat and moisture fluxes, respectively, with vertical velocity using rapid response sensors at frequencies typically equal to or greater than 10 Hz*" (**Page 7, Lines 147- 148**).

6. P. 7, line 163: see major comment 2)

**Response**: Thank you for your help and suggestions in improving our manuscript. We used average $ET_{WB}$ over multiple years instead of in one year for regional-scale validation (**Page 17, Lines 320-326**). $\Delta S$ over multiple years can be ignored (Liu et al., 2016; Senay et al., 2011) (**Page 8, Lines 174-176**).

7. P. 8, line 180: RMSE is not suited to describe model bias.

**Response**: Thank you for your help and suggestions in improving our manuscript. RMSE is the most indicator to describe the model accuracy. We rephased this sentence by replacing "*bias*" with "*the performance of the model*" (**Page 8, Line 188**).

8. P. 9, line 225: The authors conclude that 'ET_SEBAL is relatively reliable for daily-scale Application'. I am wondering how they justify that statement and if they have some references to define what a relatively reliable model performance is.

**Response**: Thank you for your help and suggestions in improving our manuscript. We deleted this sentence in this location (**Page 10, Line 230**), and added it in Section 4.1: "*Overall, the SEBAL ET showed an acceptable performance in China by comparing previous studies.*" (**Page 20, Line 374**)

9. P. 11, line 240: see major comment 5)

**Response**: Thank you for your help and suggestions in improving our manuscript. In this study,

the eight towers basically cover most of climate zones, ecosystems and terrain in China, and the observed period is long-time-series (each tower both have more than 1000 samples with total available samples of 9896). Therefore, the validation of MOD16 in this study is convincingness to some extent. The details were described in the response of major comment 5.

10. P.21, line 390: The authors state that a decrease in surface temperature corresponds to a reduced evapotranspiration. I think this needs rephrasing because in general low surface temperature at similar meteorological forcing would indicate that more of the available energy is dissipated via ET than sensible heat flux.

**Response**: Thank you for your help and suggestions in improving our manuscript. We rephased this sentence to make it clearer: "*Moreover, it should be noted, due to the decrease of surface available radiation energy which was caused by cloud cover, the ET (both actual and modeled value) is also less than that of nearest date (Cheng et al., 2020)*" (**Page 21, Lines 394-395**)

11. P. 22, line 428: I am wondering why the authors decided for the described upscaling method, if they explain in this section why other methods would be preferable.

**Response**: Thank you for your help and suggestions in improving our manuscript. In this study, we would like to discuss the sources of ET estimation errors, upscaling method may be one of the sources of errors. The upscaling method which was used in this study (constant evaporative fraction) will cause a negative bias of 10–20% in the estimation of daily ET (Delogu et al., 2012; Ryu et al., 2012; Van Niel et al., 2012), be that as it may, this method has also been widely used. Moreover, although other methods have been proposed, however, they also have a certain error and uncertainties (Gentine et al. 2007). (**Pages 22-23, Lines 434-445**)

12. P. 23, line 444: see major comment 4)

**Response**: Thank you for your help and suggestions in improving our manuscript. Due to the limited of flux tower data, we could not study the instantaneous energy component in this paper, and this object will be conducted in our follow-up research. The details were described in the response of major comment 4.

13. P. 23, line 454: What do the authors mean by 'a low domain size' in this sentence?

**Response**: Thank you for your help and suggestions in improving our manuscript. The domain size is defined here as the actual size of the modeling domain/satellite imagery being used. (**Page 23, Lines 457-458**)

14. P. 26, line 538: the use of the arrows for indicating up- and downwelling radiation is inconsistent between equations and the text.

**Response**: Thank you for your help and suggestions in improving our manuscript. We modified them to $R_{s\_down}$, $R_{l\_up}$, and $R_{l\_down}$. (**Page 27, Lines 550-556,**)

15. P. 27: line equation (12) is true for neutral conditions only.

**Response**: Thank you for your help and suggestions in improving our manuscript. The computation of $r_a$ is modified based on Eq. 30, when it is not neutral conditions, the Eq. 30 will be adjusted by Monin–Obkhov length which could judge whether it is stable or unstable conditions. The Eq. 12 was used at the first time of the loop only, as the neutral conditions was assumed. The Eqs. 19 – 30 could describe this process (**Pages 29-30, Lines 610-626**).

16. P. 28, line 580: see major comment 1)

**Response**: Thank you for your help and suggestions in improving our manuscript. We supplemented parts of content to compare the different domain sizes' performance in ET estimation (**Fig. 12, Page 24, Lines 475**). The details were described in the response of major comment 1.

17. P. 28: some of the equations (20) to (28) are redundant.

**Response**: Thank you for your help and suggestions in improving our manuscript. Eqs. 20-30 clearly describe the process of $r_a$ computation at different conditions (neutral, stable or unstable), so we prefer to remain it.

18. Zenodo homepage: The authors state 'The products were evaluated using the eight flux towers observation data for point validation and water balance method for regional validation and showed

R value of 0.79 and 0.88, respectively, which indicated the products have a great Performance'. In my opinion a rRMSE of > 40 % might maybe not indicate great performance.

**Response**: Thank you for your help and suggestions in improving our manuscript. As described in Section 4.1, SEBAL ET showed a comparable performance in China with previous studies, and it is better than MOD16 products. "great performance" may not very suit, so this sentence was changed to "*The products were evaluated using the eight flux towers observation data for point validation and water balance method for regional validation and showed R value of 0.79 and 0.98, respectively, which indicated the products have a better performance than MOD16 products which have been widely used*". In general, the RMSE (0.92 mm/d for point scale and 48.99 mm/year for regional scale) and rRMSE (42.04% for point scale and 13.57% for regional scale) of SEBAL ET are an acceptable accuracy at current, which due to the validation methods still have errors, e.g., EC tower have a typical error of 5-20% in ET observation, be that as it may, EC tower is still a widely used in situ measuring method.

19. Zenodo homepage: the coordinate system should be stated in the text

**Response**: Thank you for your help and suggestions in improving our manuscript. The coordinate system is GCS_WGS_1984, and have added in the statement of Zenodo homepage:

"*The dataset named SEBAL evapotranspiration in China (SEBAL ET) characterized the daily evapotranspiration (in millimeter) of vegetation in China from 2001 to 2018, the spatial resolution is 1 km × 1 km and the temporal resolution is 1 day with the coordinate system of GCS_WGS_1984. The products were generated using Surface Energy Balance Algorithm of Land (SEBAL) and multi-sources remote sensing data, including MOD43A1 daily surface albedo, MOD11A1 daily surface temperature and MOD13 vegetation indices (obtained from NASA: https://ladsweb.modaps.eosdis.nasa.gov/search/), the meteorological data obtained from GMAO (https://gmao.gsfc.nasa.gov/research/highlights/2013-2015.php), the input variables were all aggregated of resampled to 1 km × 1km. The products were evaluated using the eight flux towers observation data for point validation and water balance method for regional validation and showed R value of 0.79 and 0.98, respectively, which indicated the products have a better performance than MOD16 products which have been widely used. SEBAL ET can be used for several geoscience studies, especially for global change, water resources management and*

*agricultural drought monitoring, etc.*"

---

## Author Response (AR3)

**Ref.: Dr. No. ESSD-2020-345**

**Title**: Long time series of daily evapotranspiration in China based on the SEBAL model and multisource images and validation

**Author**: Minghan Cheng, Xiyun Jiao, Binbin Li, Xun Yu, Mingchao Shao, Xiuliang Jin

**Research Paper**

**Earth System Science Data**

**Cover letter**

**Dear Editor and Reviewers**

I am submitting here a manuscript entitled "*Long time series of daily evapotranspiration in China based on the SEBAL model and multisource images and validation*". We submitted this manuscript in November 2020. Three reviewers gave us good advices. First we would like to thank the reviewers for their constructive and helpful suggestions and improvements to our manuscript (**ESSD-2020-345**). We revised the manuscript by following the suggestions of the reviewers. Our response to each suggestion or comment are given one by one in the following Pages of this letter. For details, please refer to the responses as follows (Reviewer comments are in black font, responses are in blue or red font)

Looking forward to your favorable decision.

Thanks too much.

With best regards,

Minghan Cheng and co-authors

**Responses to Reviewers**

**Reviewer 3#:**

The authors addressed most of my concerns. Below I list a few further comments:

**Comment 1:**

The authors reply to my comment concerning the EC measurement principle: *We have rewritten this sentence by referring the paper of Wang et al. (2012): The eddy covariance method measures λET from the covariance of the heat and moisture fluxes, respectively, with vertical velocity using rapid response sensors at frequencies typically equal to or greater than 10 Hz (Page 7, Lines 147- 148).* This is again wrong. And the authors use the paper Wang and Dickinson 2012 as a reference, but cite it incorrectly. The original sentence from the paper reads: *The EC technique measures **H and lE** from the covariance of the heat and moisture fluxes, respectively, with vertical velocity using rapid response sensors at frequencies typically equal to or greater than 10 Hz.* The covariance between heat fluxes and vertical wind velocity defines the sensible heat flux, while the covariance between the moisture flux and the vertical wind velocity defines the latent heat flux. That the authors decided to cite this sentence only partially (leaving out the sensible heat flux) is very critical in my view. This repeated wrong definition of the EC method, makes me doubt the authors understanding of the EC theory.

**Response:** Thank you for your help and suggestions in improving our manuscript. Regarding to the definition of latent heat flux measuring by EC tower, we revised this sentence again: *'The eddy covariance method measures λET from the covariance between moisture fluxes and vertical wind velocity using rapid response sensors at frequencies typically equal to or greater than 10 Hz.'* (Lines 147-148, Page 7).

**Comment 2:**

The authors replied very detailedly to my comment on the differentiation into different climate zones, terrain types etc. by referencing other papers that used a similar number of EC stations for assessing model performance. In my opinion, this is still not convincing since e.g., the class cropland includes a single site while croplands exhibit very different ET rates and

dynamics based on the cultivated crop. The same will hold true e.g., for climate types. In my opinion the authors should at least state that the low number of stations per class (land cover, terrain type etc.) might reduce the validity of these findings.

**Response:** Thank you for your help and suggestions in improving our manuscript. Although different crop types show different ET, this EC observations were used to verify the accuracy difference of SEBAL among different land surface types. Moreover, we classify cropland into one category because the difference between different crops, such as wheat and maize, is relatively small compared with cropland and woodland or cropland and grassland.

The number of flux towers could determine the reliability of the verification results to some extent. We agree with the reviewers. Therefore, we added a paragraph in the discussion:*In general, more ET sites could improve the reliability of the validation process. However, limited number of EC towers in this study also caused uncertainties in SEBAL ET validation. Although the validation of SEBAL was conducted in different situations (e.g., different climate zones and land cover types), and it should be noted that several situations only have one sites can be used, e.g., cropland. Therefore, only one climate zone of cropland ET can be validated, and other situations were not considered. Moreover, in this study, the different classes of land cover types, climate zones, elevation and seasons were considered for SEBAL validation by referred to previous studies (Kim et al., 2012; Ramoelo et al., 2014). However, there are still several situations may need to be considered. For example, whether the difference of SEBAL accuracy was existed in different years (Velpuri et al., 2013) and different satellite sensors (Long et al., 2011). Overall, more ground-based sites should be used to get more reliable results of SEBAL validation in the follow-up research.* (Lines 432-440, Page 22)

**Comment 3:**

The equations (26) to (28) are still redundant.

**Response:** Thank you for your help and suggestions in improving our manuscript. We have deleted the redundant equation (Eqs. 26 - 28) and only remained one equation which could contain the original three equations (Line 629, Page 29).

**Comment 4:**

The manuscript still needs English proofreading.

**Response:** Thank you for your help and suggestions in improving our manuscript. We have checked the English writing and grammar; besides, the manuscript has also been checked by a native-speaker.

**Editor#:**

as referee #3 is generally happy that you revised the manuscript according to most of the comments, but still raises two important points, I would encourage you to address these (and the other comments) thoroughly.

Major points:

**Comment 1:**

- Please ensure that you cite the EC literature correctly and provide a comprehensible understanding of EC theory. This will help the readers to best understand the suitability of the methodology and resulting dataset.

**Response:** Thank you for your help and suggestions in improving our manuscript. We have cited the EC literature correctly and provided a comprehensible understanding of EC theory according to your and reviewers' comments. The revised sentence is as follows: *'The eddy covariance method measures λET from the covariance between moisture fluxes and vertical wind velocity using rapid response sensors at frequencies typically equal to or greater than 10 Hz.'* (Lines 147-148, Page 7).

**Comment 2:**

- The "Comment 2" requires some more explanation and critical discussion of methodology on your part. If the referee is not convinced about the suitability of the approach, other readers might also be skeptical. You should address this by offering the limitations of the approach, clear description, possible uncertainties, and a critical discussion of the implications for the model validation.

**Response:** Thank you for your help and suggestions in improving our manuscript. According to the suggestion of the Reviewer, we supplemented a paragraph in the Discussion (Lines 432-440, Page 22).

---

## Author Response (AR4)

**Ref.: Dr. No. ESSD-2020-345**

**Title**: Long time series of daily evapotranspiration in China based on the SEBAL model and multisource images and validation

**Author**: Minghan Cheng, Xiyun Jiao, Binbin Li, Xun Yu, Mingchao Shao, Xiuliang Jin

**Research Paper**

**Earth System Science Data**

**Cover letter**

**Dear Editor:**

I am submitting here a manuscript entitled "*Long time series of daily evapotranspiration in China based on the SEBAL model and multisource images and validation*" (**ESSD-2020-345**). The manuscript has been revised three rounds by referred to three Reviewers' good suggestions. First we would like to thank the reviewers for their constructive and helpful suggestions and improvements to our manuscript. Our response to each suggestion or comment are given one by one in the following Pages of this letter. For details, please refer to the responses as follows (Reviewer or editor comments are in black font, responses are in blue or red font)

Looking forward to your favorable decision.

Thanks too much.

With best regards,

Minghan Cheng and co-authors

**Editor comments:**

Regarding the paragraph you added as a response to comment #2 (list in the follow), about the validation of your data product with limited EC tower availability, I'm afraid the paragraph you added is not easily understood and also does not completely address the concerns of the referee.

Could you please re-write the paragraph (and also, please consult a native speaker again for the English phrasing, some of it could use more clarity), so that you actually discuss the validation basis for your different ET classes, and give some sort of uncertainty estimation. Please take special care to discuss the implication of having only one ground truth EC tower for a class. This is relevant for people who want to use your dataset, it should be clear which parts of the classification have a good validation basis and which should be used with great caution.

**The comment 2 of Reviewer #3:**

*The authors replied very detailly to my comment on the differentiation into different climate zones, terrain types etc. by referencing other papers that used a similar number of EC stations for assessing model performance. In my opinion, this is still not convincing since e.g., the class cropland includes a single site while croplands exhibit very different ET rates and dynamics based on the cultivated crop. The same will hold true e.g., for climate types. In my opinion the authors should at least state that the low number of stations per class (land cover, terrain type etc.) might reduce the validity of these findings.*

**Response:** Thank you for your help and suggestions in improving our manuscript. We have rewritten this paragraph (Lines 432-446, Pages 22-23). The potential bias in the findings was acknowledged and analyzed. Compared to classes which had multiple validation sites, the representativeness of the evaluation was inevitably compromised in cropland, tropical zone, and warm-temperate zone, which had only one site available. Nevertheless, this deficiency was alleviated in this study by incorporating the long-time-series data. The details are as follows:

*'Although a comprehensive evaluation of SEBAL ET over different classes was conducted in this study, users should be aware of the uncertainties due to the limited number of validation sites in some classes. For example, only one site was available for the*

*evaluation over cropland. Because this cropland flux tower site was set in plain and warm-temperate zone, the accuracy may only represent the data quality of cropland ET in the warm-temperate plain zone, but not other regions. Nevertheless, long-time-series data were obtained from this site which covered different seasons and different crop types. Employing these hundreds of samples in the validation could remedy the single-site insufficiency to a certain extent. Similarly, only one site was found in the validation over two other classes, i.e., tropical zone and warm-temperate zone. Long-time-series data were also incorporated to enhance the representativeness of the single site. Regarding the other classes, two or more sites were used which will lead to more reliable results. Compared to pervious studies (Aguilar et al., 2018; Hu et al., 2015; Ramoelo et al., 2014; Yang et al., 2017), a larger number of validation samples (flux tower sites) were used in this study, indicating that the findings were reliable. Additionally, although the validation of SEBAL ET in this study followed the literature (Kim et al., 2012; Ramoelo et al., 2014) and considered different land cover types, climate zones, elevation and seasons, several more situations may need to be considered. For example, whether the SEBAL accuracy was different across years (Velpuri et al., 2013) and satellite sensors (Long et al., 2011). Overall, with the increasing number of flux towers set up in China, more reliable and comprehensive validation of SEBAL ET can be conducted in the follow-up research.'*